

# Linking Gulf Stream Air-Sea Interactions to the exceptional blocking episode in February 2019: A Lagrangian Perspective

Marta Wenta[1], Christian M. Grams[1], Lukas Papritz[2], and Marc Federer[2]

[1]Institute of Meteorology and Climate Research, Department Troposphere Research (IMK-TRO), Karlsruhe Institute of Technology (KIT), Karlsruhe, Germany
[2]Institute for Atmospheric and Climate Science, ETH Zürich, Zurich, Switzerland

**Correspondence:** Marta Wenta (marta.wenta@kit.edu)

**Abstract.**

The development of atmospheric blocks over the North Atlantic European region can lead to extreme weather events like heatwaves or cold air outbreaks. Despite their potential severe impact on surface weather, the correct prediction of blocking lifecycles remains a key challenge in current numerical weather prediction (NWP) models. Increasing evidence suggests that latent heat release in cyclones, the advection of cold air (cold air outbreaks, CAOs) from the Arctic over the North Atlantic, and associated air-sea interactions over the Gulf Stream are key processes responsible for the onset, maintenance, and persistence of such flow regimes. In order to establish how air mass transformations over the ocean, and in particular over the Gulf Stream, affect the large-scale flow, we focus on an episode between 20 and 27 of February 2019, when a quasi-stationary upper-level ridge established over western Europe accompanied by an intensified storm track in the Northwestern North Atlantic. During that time a record-breaking warm spell occurred over Western Europe bringing temperatures above 20°C to the United Kingdom, the Netherlands, and Northern France. The event was preceded and accompanied by the development of several, rapidly intensifying cyclones originating in the Gulf Stream region and traversing the North Atlantic. To explore the mechanistic linkage between the formation of this block and air-sea interactions over the Gulf Stream, we adopt a Lagrangian perspective, using backward and forward kinematic trajectories. This allows us to study the pathways and transformations of air masses forming the upper-level potential vorticity anomaly and interacting with the ocean front. We establish that more than one-fifth of these air masses interact with the Gulf Stream in the lower troposphere, experiencing intense heating and moistening over the region, due to the frequent occurrence of CAOs behind the cold front of the cyclones. Trajectories moistened within the cold sector of one cyclone, later ascent into the upper troposphere with the ascending air stream of a consecutive cyclone, fueled by the strong surface fluxes. These findings highlight the importance of CAOs in the Gulf Stream region with their intense coupling between the ocean and atmosphere for blocking development, and provide a mechanistic pathway linking air-sea interactions in the lower troposphere and the upper-level flow.

## 1 Introduction

Atmospheric blocks are quasi-stationary anticyclonic circulation anomalies disrupting the eastward propagation of synoptic weather systems. The associated high-pressure system can dominate the weather over a particular location for an extended





period of time, from several days to weeks (Wazneh et al., 2021) and lead to the development of extreme weather, like cold
surges (e.g. de'Donato et al., 2013; Demirtaş, 2017; Pang et al., 2020; Zhuo et al., 2022) and heatwaves (e.g. Grumm, 2011;
Barriopedro et al., 2011; Spensberger et al., 2020; Dae et al., 2022; Kautz et al., 2022) with significant socio-economic im-
pacts. Despite ongoing development and increasing resolution of numerical climate and weather prediction models, the correct
prediction of those quasi-stationary weather patterns still poses a challenge (Matsueda and Palmer, 2018; Ferranti et al., 2018;
Grams et al., 2018; Büeler et al., 2021).

The dynamics of cyclones and blocking anticyclones are mutually linked with the position and tilt of the upper-tropospheric
jet. The crucial role of cyclones for the formation and maintenance of the blocks has been established by multiple studies (e.g.
Colucci, 1985; Colucci and Alberta, 1996; Lupo and Smith, 1995; Nakamura and Wallace, 1993; Mullen, 1987; Yamazaki and
Itoh, 2009). The development of cyclones results in the cross-isentropic ascent of air from the lower to the upper troposphere, in
the so-called Warm Conveyor Belt (WCB; Wernli and Davies, 1997; Madonna et al., 2014; Pfahl et al., 2014). Condensation and
resulting latent heat release during the ascent are critical for both cyclone intensification, through the production of potential
vorticity (PV) below the level of maximum heating (Binder et al., 2016; Reed et al., 1992; Čampa and Wernli, 2012), and
growth of the upper-level ridge, due to the destruction of PV above the level of maximum heating (Methven, 2015; Madonna
et al., 2014; Joos and R.M.Forbes, 2016; Grams et al., 2011). The injection of low PV air into the upper troposphere together
with diabatically enhanced divergent outflow amplifies and reinforces the upper-tropospheric ridge (Grams et al., 2011; Teubler
and Riemer, 2016; Grams and Archambault, 2016; Steinfeld and Pfahl, 2019). Diabatic processes, as recently quantified by
Pfahl et al. (2015); Steinfeld and Pfahl (2019); Steinfeld et al. (2020); Yamamoto et al. (2021) are in many cases essential for
the development of blocks in the North Atlantic-European region. In fact, recent studies indicate that the duration, strength, and
possibly even formation of the block are influenced by latent heat release in the ascending airstreams (Steinfeld et al., 2020;
Pfahl et al., 2015).

The key role of moist dynamics in blocking formation and development suggests that there exists a relationship between up-
stream, lower-tropospheric processes and the formation of the upper-level, quasi-stationary ridge. The air masses that undergo
latent heat release during the ascent in the warm sector of the cyclone need to first pass through the region of intense surface
evaporation to pick up a sufficient amount of moisture. Recent studies suggest that during winter the moisture source locations
of cyclone precipitation are fairly local and over the ocean (Pfahl et al., 2014; Papritz et al., 2021). In the North Atlantic,
the most intense evaporation events are associated with the Gulf Stream (Aemisegger and Papritz, 2018). The propagation of
cyclones across the Gulf Stream region provides conditions for large surface latent and sensible heat fluxes (Tilinina et al.,
2018; Moore and Renfrew, 2002), due to the development of CAOs and the descent of dry air in the cold sector (Vanniére
et al., 2017; Raveh-Rubin, 2017; Aemisegger and Papritz, 2018). The warm waters of the Gulf Stream have been identified by
Papritz et al. (2021) as a primary moisture source for cyclone-related precipitation in the North Atlantic. Papritz et al. (2021)
demonstrated also that air masses moistened and heated in the cold sector of one cyclone are then brought into the warm sector
of the consecutive cyclone through a cyclone relative flow, called feeder airstream (Dacre et al., 2019). Such cyclone-cyclone
interaction has also been identified and labeled as a 'hand-over' mechanism by Sodemann and Stohl (2013). Moreover, intense
turbulent heat fluxes during CAO events also play a crucial role in the restoration of baroclinicity in the lower troposphere





(Papritz and Spengler, 2015; Vanniére et al., 2017), and precondition the atmosphere for the development of consecutive low-pressure systems (Tilinina et al., 2018; Papritz et al., 2021; Vannière et al., 2017). In consequence, CAOs can regulate cyclone formation and strength and hence potentially affect downstream large-scale dynamics (Papritz and Grams, 2018). However, the pathway of CAO influence on the upper-level flow is unclear.

Previous studies demonstrated that intense heat transfer in the regions of western boundary currents influences the posi-
tion of the storm tracks (Kwon et al., 2010; Shaw et al., 2016) and plays an important role in the upper-level jet variability (Nakamura et al., 2008). In fact, Kwon et al. (2010) found in their modeling study that the absence of the Gulf Stream sea surface temperature (SST) gradient results in a reduced frequency of blocks downstream. Furthermore, O'Reilly et al. (2017) determined that wintertime poleward displacements of the jet stream are preceded by high eddy heat fluxes over the Gulf Stream and western North Atlantic. The mechanism behind those displacements is explained by Novak et al. (2015) and Kwon
et al. (2020), who showed that the shift of the upper-level flow is caused by the northward shift of eddy heat flux in the lower troposphere. An increasing number of studies also indicate that the Gulf Stream region might serve as a moisture source for air masses ascending into the blocking regions. Yamamoto et al. (2021), using a 31-year climatology of backward air trajectories started from the upper-level North Atlantic-European blocks, found that the Atlantic basin provides most of the moisture for the moist air masses ascending into the block. Moreover, they established that trajectories that gather moisture from the ocean
follow the path of the Gulf Stream and identified the region of the SST gradient in the western North Atlantic as the region where trajectories ascent to the upper troposphere. Those results are also in agreement with the findings of Pfahl et al. (2014), who determined that moisture supplies for WCBs are collocated with the regions of intense ocean evaporation in the western North Atlantic.

Throughout the literature, researchers have established the importance of ocean-atmosphere coupling over the Gulf Stream
and its relevance for downstream large-scale dynamics (e.g. Vanniére et al., 2017; Sheldon et al., 2017; Papritz and Spengler, 2015). However, the scientific community has yet to gain a clear understanding of the physical pathway through which signals from individual processes in the marine boundary layer are conveyed to the large-scale circulation (Czaja et al., 2019). In this study, we propose a possible explanation for this missing mechanistic link by conducting a case study of European Blocking from February 2019. This event brought record-breaking heatwaves to western Europe and was accompanied by a series of
upstream, rapidly intensifying cyclones. We explore the relationship between air-sea interactions over the Gulf Stream region and the formation of an upper-level ridge over western Europe from both backward and forward Lagrangian perspectives in a synoptic study. We structure the paper as follows: First, we provide a detailed description of the data and methods, including trajectory calculations (Section 2). Second, we introduce the European Blocking case study of February 2019 (Section 3.1). In the following section, we describe in detail the results of our analysis, which focuses on the connection between the Gulf
Stream and the development of an upper-level, quasi-persistent ridge (Section 3.2). Then, we analyze the moisture sources and transport paths of the air ascending into the block (Section 3.3), and explain the variations of negative PV in the lower troposphere (Section 3.5.1) and their possible relevance for blocking development. Finally, we carry out a discussion of the results and draw final conclusions, synthesizing the results into a mechanistic link between air-sea interactions over the Gulf Stream, the development of cyclones, and the formation of the European Blocking event in February 2019.



## 2    Methodolody

### 2.1    Data

#### 2.1.1    ERA5 reanalysis

The calculation of kinematic trajectories and the analyses presented in this study are based on European Center for Medium-Range Weather Forecasts (ECMWF) reanalysis - ERA5 (Hersbach et al., 2020). For the most of the study, we use reanalysis data at the 3-hourly temporal resolution, interpolated on a $0.5°x0.5°$ horizontal grid. In addition, we employ ERA5 data with a higher temporal resolution of one hour for the investigation of cyclone tracks (Section 2.1.2), the vertical and horizontal distribution of negative potential vorticity, and the cloud liquid water content (Section 3.5.1). We chose the lowest 98 sigma-pressure vertical levels out of a total of 137 available levels for our investigation, covering the pressure range from ∼26 hPa to the surface. The analyzed data covers the period from 10 to 28 February 2019.

#### 2.1.2    Cyclone dataset

The cyclone tracks are obtained using the method of Sprenger (2017) and Wernli and Schwierz (2006), based on the identification and tracking of the sea level pressure (SLP) minima, defined as the grid points with SLP value lower than at the eight neighboring grid points. In addition, the cyclone extent is determined by the outermost closed SLP contour surrounding the identified SLP minima. The tracking algorithm is applied to hourly fields of SLP from ERA5 reanalysis. Rapidly intensifying cyclones are identified using the criterion of Sanders and Gyakum (1980) of a central pressure drop of at least 24 hPa within 24 hours. Complementary to the cyclone identification we use the surface sensible heat flux (SSHF) and the potential vorticity (PV) in the lowest model level to distinguish the cold sectors of the cyclones, following Vannière et al. (2017). In contrast to Vannière et al. (2017) we use a threshold of the SSHF higher than 0 $W/m^2$ and the PV lower than 0.1 PVU (1 PVU = $10^{-6}Kkg^{-1}m^2s^{-1}$). In their work Vannière et al. (2017) indicate that for some instances different thresholds can be more fitting. The thresholds of SSHF>0 $W/m^2$ and PV<0.1 PVU were chosen based on the visual analysis of cold sectors' spatial distribution.

#### 2.1.3    Identification of the block and upper-troposphere negative potential vorticity anomalies.

The European Block in February 2019 is identified using the year-round weather regime definition of Grams et al. (2017) for the North Atlantic-European region. The Block is characterized by a positive geopotential height anomaly over the eastern North Atlantic and Europe and a negative geopotential height anomaly upstream over Greenland. The methodology for identifying specific weather regimes is described in detail in Grams et al. (2017) and Hauser et al. (2022).

The formation of the atmospheric block in the Euro-Atlantic region is associated with the poleward advection of low PV air. The accumulation of low PV air in the upper troposphere leads to the development of negative potential vorticity anomalies (NPVA; Teubler and Riemer, 2016), which amplify the upper-level ridge. In our study, we use the method of Hauser et al. (2022) to identify NPVAs in the ERA5 dataset. First, the deviations of PV from a 30-day running mean climatology (1979-





2019) centered on the day of interest are calculated. Then, vertical averages of obtained values between 500 and 150 hPa are computed and labeled as NPVA objects if they fall below the threshold of -0.8 PVU. In the next step, a quasi-Lagrangian framework is employed to follow the evolution of NPVAs and assign them to the lifecycle of the European Block in February 2019. PVAs are assigned to active weather regimes based on their spatial overlap with a predefined regime mask. The mask is

defined as the area where weather regime pattern values are below -0.3 PVU. If a PVA overlaps with the mask by at least 10% during an active regime, it is attributed to that specific regime life cycle. Sometimes, multiple PVAs may contribute to a single regime. The weather regime PV pattern is the average of the low-frequency PVAs vertically averaged between 500 and 150 hPa during active days of the cycle.

The formation of the studied block was related to one major NPVA, and another minor NPVA that formed on 23 February

over Greenland (Fig.1). The major NPVA formed 10 days prior to blocking onset over the North Pacific and started to strengthen a few days before the block onset when it propagated into the North Atlantic. For the purpose of the present study, we neglect the NPVAs lifecycle prior to their arrival into the North Atlantic region.

### 2.1.4 Identification of cold air outbreaks

Cold air outbreaks (CAO) in the ERA5 dataset are identified using the method of Papritz et al. (2015). First, the air-sea

potential temperature difference between $\theta_{SST} - \theta_{850}$ is calculated, where $\theta_{SST}$ denotes sea surface potential temperature and $\theta_{850}$ air potential temperature at 850 hPa. The reference pressure $p_0 =$1000 hPa is used for the calculation of surface potential temperature. In agreement with Papritz et al. (2015), we require the $\theta_{SST} - \theta_{850}$ over the ocean to exceed 0 K to identify the CAO events.

## 2.2 Trajectory datasets

The LAGRANTO analysis tool (LAGRANTO Sprenger and Wernli, 2015) is employed to calculate kinematic trajectories, using three-dimensional wind on model levels from the ERA5 dataset described above. Output positions of trajectories are available in 3-hourly intervals and the following variables are traced along the trajectories: temperature, specific humidity, potential vorticity, surface pressure, surface latent heat flux (SLHF), surface sensible heat flux (SSHF), boundary layer height and sea surface temperature (SST). Two principal trajectory datasets are compiled, labeled according to the starting regions

as NPVA and GS trajectories (Tab.1). Those datasets are further divided into subsets that serve to illustrate the connections between the block and air-sea interactions over the Gulf Stream as well as the properties of the trajectories. All collections of trajectories are listed in Tab.1 and are discussed below.

### 2.2.1 NPVA Trajectories

The first principal trajectory dataset comprises trajectories started from the upper-level NPVA objects (Section 2.1.3) every 3

hours between 20 February 09:00 UTC and 28 February 12:00. The 10-day kinematic backward trajectories are initiated from





| Name of the dataset | Starting area | Duration backward | Duration forward | Characteristics |
|---|---|---|---|---|
| **NPVA trajectories** | | | | **Ascent of 500 hPa 10 days prior to the arrival in NPVA.** |
| NPVA GS trajectories | | | | Interact with the ABL over the Gulf Stream. |
| NPVA CAO1 GS trajectories | | | | Undergo CAO ($\theta_{SST} - \theta$>0) |
| NPVA CAO2 GS trajectories | | | | Undergo CAO ($\theta_{SST} - \theta$>2) |
| NPVA WCB GS trajectories | Upper-troposphere NPVA. | 10 days | - | Fulfil the WCB criterium of 600 hPa/48 h ascent. |
| NPVA DI GS trajectories | | | | Fulfil the DI criterium of 400 hPa/48 h descent. |
| NPVA DH GS trajectories | | | | Diabatically heated within 3 days after initialization. |
| NPVA nonGS trajectories | | | | Do not interact with the ABL over the Gulf Stream. |
| **GS trajectories** | | | | **Ascent of 500 hPa within 10 days forward from the initialisation.** |
| GS NPVA trajectories | | | | Ascent into the upper-troposphere NPVA objects. |
| GS CAO1 NPVA trajectories | | | | Undergo CAO ($\theta_{SST} - \theta$>0) |
| GS CAO2 NPVA trajectories | | | | Undergo CAO ($\theta_{SST} - \theta$>2) |
| GS WCB NPVA trajectories | The Gulf Stream mask. | 10 days | 10 days | Fulfil the WCB criterium of 600 hPa/48 h ascent. |
| GS DI NPVA trajectories | | | | Fulfil the DI criterium of 400 hPa/48 h descent. |
| GS DH NPVA trajectories | | | | Diabatically heated within 3 days after initialization. |
| GS NPVA negPV trajectories | | | | Negative PV in the lower troposphere. |
| GS NPVA posPV trajectories | | | | Positive PV in the lower troposphere. |

**Table 1.** Overview of the trajectory subsets used in the study. The darker, bold rows indicate initial trajectory setups, while lightly shaded ones refer to the selection of trajectories based on their relation to the Gulf Stream (GS) mask or negative potential vorticity anomaly objects (NPVA). White rows refer to the further division of selected trajectories into different subsets and airstreams.

an equidistant grid of $\Delta x$=100 km and $\Delta y$=25 hPa vertically between 500 and 150 hPa within both NPVAs (Fig.1, Section 2.1.3).

In the consecutive analysis, the obtained trajectory dataset is refined as we apply additional filtering and selection criteria. First, to select only the ascending trajectories, we require trajectories to experience a pressure decrease of 500 hPa within 10 days prior to the arrival in the upper-level NPVA, hence the air parcel can ascend at any time and any rate. The threshold of 500 hPa is chosen to ensure that the trajectory has ascended all the way from the lower troposphere, enabling the analysis of ocean influence on the ascending air. Approximately 53% of the trajectories in the initial dataset experience such an ascent of 500 hPa before their arrival into the upper-level NPVA. Second, to avoid the possibility of trajectory double-counting, we detect those that remain for two consecutive time steps within the starting grid of the NPVA and remove them from the dataset. This filtering technique removes approximately 10% of trajectories. The remaining trajectories will be referred to throughout the following analysis as 'NPVA trajectories' (Tab.1).

### 2.2.2 NPVA GS Trajectories

Taking into account the importance of the Gulf Stream for the selected study, we create an additional subset of trajectories consisting of only those NPVA trajectories that have passed over the Gulf Stream in the lower troposphere. We define the boundary for the lower troposphere as 800 hPa, commonly used as an upper limit of the WCB inflow (e.g. Binder et al., 2020). The region of the Gulf Stream (GS masks) is defined for every 3-hourly timestep of the ERA5 dataset for February 2019 using





the following steps: (i) first, the horizontal gradient of the SST is identified in both west-east and north-south directions, (ii) a threshold of $|\nabla SST| > 2\text{K}$ is applied to extract the area of the Gulf Stream SST front, (iii) a buffer of 100 km is added to the identified gradient, creating a continuous region.

In the following, we refer to those trajectories as 'NPVA GS trajectories' (Tab.1). The rest of the trajectories, i.e. those that did not interact with the ABL over the Gulf Stream, are labeled as 'NPVA nonGS trajectories'.

### 2.2.3 GS Trajectories

The second principal dataset consists of trajectories that are started from the lower troposphere over the region of the Gulf Stream, defined using the same masks as for the selection of NPVA GS trajectories (see Section 2.2.2). Those trajectories are
run both backward and forward in time for 10 days, in total having a duration of 20 days with time 0 indicating their location over the Gulf Stream SST front region. The spatial resolution of the equidistant starting grid is $\Delta 50$ km in the horizontal and $\Delta y=25$ hPa in the vertical ranging from 700 to 1000 hPa. Similar to the previous setup, only those trajectories that experience a change of pressure of 500 hPa within 10 days from the start forward in time are selected. This criterion is applied as the goal of the study is to establish the connection between air-sea interactions over the Gulf Stream and upper troposphere dynamics.
Approximately 43% of trajectories of the original dataset meet this criterion. Furthermore, to avoid double-counting, a filtering criterion is applied to eliminate trajectories remaining within the starting region for the first two time steps, leading to the reduction of the total number of trajectories by $\sim$15%. The acquired dataset is labeled as 'GS trajectories' (Tab.1).

### 2.2.4 GS NPVA Trajectories

In a similar manner as in the case of NPVA GS trajectories, we select a subset of the GS trajectories, that at some point within
10 days from leaving the Gulf Stream region end up in the NPVAs (Section 2.1.3) associated with the analyzed European Block (GS NPVA trajectories, Tab. 1).

### 2.2.5 Further trajectory subsets

In each collection of the NPVA and GS trajectories we additionally identify the following air streams (Tab.1):

– CAO - to determine if a trajectory is a part of a CAO, we consider $\theta_{SST} - \theta$, where $\theta$ is the air parcel potential temperature
(see Papritz and Spengler (2017). We use two different thresholds to identify CAOs, 0 K(CAO1) and 2 K (CAO2), to distinguish between weaker and stronger CAOs.

– DH - trajectories that experienced diabatic heating of $\Delta\theta \geqslant 2$ K within 3 days before their arrival in the block, following Pfahl et al. (2015).

– WCB - trajectories fulfilling the warm conveyor belt criterion of 600 hPa ascent within 48h or less in, line with Madonna
et al. (2014), in any time window within the 10 days backward calculation for NPVA trajectories and 10 days back-





ward/forward for GS trajectories, with the additional rule for GS trajectories that the WCB ascent has to end within the forward part of the trajectory.

– DI - trajectories fulfilling the dry intrusion criterion of 400 hPa descent within 48h following the definition of Raveh-Rubin (2017), in any time window within the 10 days backward calculation for NPVA trajectories and 10 days forward/backward for GS trajectories, with the additional rule for GS trajectories that the dry intrusion's descent has to start in the upper troposphere before the air parcel reaches the Gulf Stream.

## 2.3 Moisture source identification

The method of Sodemann et al. (2008) is applied for the purpose of moisture source identification. In this approach, a specific humidity change along a trajectory is considered as an uptake if the specific humidity difference between two-time steps (difference of 3 h) exceeds 0.02 g/kg. Each uptake is given a weight based on all consecutive changes in the specific humidity along the trajectory. This means that the contribution of each uptake is adjusted by considering precipitation events en route and subsequent uptakes. This method has been widely recognized as appropriate for the identification of moisture sources and used in a number of other studies (e.g. Papritz et al., 2021; Xin et al., 2022; Jullien et al., 2020; Aemisegger and Papritz, 2018).

This method is applied to both the NPVA GS and the NPVA nonGS trajectories to identify the sources of moisture present at the start of the ascent. The use of the start of trajectory's ascent as a reference time for the moisture diagnostic allows us to identify the sources of moisture contributing to latent heat release during an air parcel's upward movement. For the purpose of this analysis, every backward trajectory (NPVA GS and NPVA nonGS, Tab.1) is extended 10 days backward from the time when the ascent started.

## 3 Results

### 3.1 The European Blocking Heatwave 2019

The European Blocking event in February 2019 lasted for about 7 days, from 20 to 27 February. The duration of this event was below the average for winter block events in the Northern Hemisphere (Wazneh et al., 2021). However, it was accompanied by record high temperatures for this month in France, the Netherlands, and the United Kingdom (Young and Galvin, 2020) with 2 m temperature anomalies in western Europe exceeding +10 °C (Fig.1). This exceptional, wintertime heatwave was linked to the formation of a quasi-stationary, upper-level ridge, which brought southerly airflow and clear skies to western Europe (Leach et al., 2021).

Temperature data from weather stations illustrate the extreme nature of this event. The highest temperature anomalies were observed on 26 and 27 February (Fig.1, h), with the record high temperature in February for the United Kingdom of 21.2 °C measured in Kew Gardens, London (Young and Galvin, 2020). Record-breaking observations were also made in Scotland (18.3°C), the Netherlands (18.9 °C), and Sweden (16.7 °C), highlighting the spatial extent of the event (Young and Galvin, 2020).



Europe has already experienced moderate winter weather prior to the blocking event. In the second part of February, the upper-level flow was repeatedly interrupted by the formation of upper-tropospheric NPVAs. Two days prior to the analyzed event, on 18 February, the west-to-east propagation of the jet stream was disrupted by the NPVA in the upper troposphere
stretching over western Europe and another over the central North Atlantic (white, dashed contours in Fig.1a). The resulting anticyclonic circulation was accompanied by south and southwesterly flow bringing warm temperatures to central Europe with anomalies exceeding 10°C (Fig.1b). Three days later, the upper-level flow was disturbed by another NPVA, extending from southern Europe to the North Atlantic and Greenland (grey shading in Fig.1c). In contrast to the NPVA object from 18 February, this new NPVA became quasi-stationary and persisted over the region for a week. Clear skies and the continued
inflow of warm air associated with the anticyclonic circulation (Leach et al., 2021) further intensified the warm spell over France, the Netherlands, and the United Kingdom (Fig.1d, f). On 23 February, the upper-level flow was further disturbed by another minor NPVA (light blue in Fig.1e), which strengthened the block and lead to its extension westward. 27 February marks the last day of the blocking event when both the major and minor NPVAs started to shrink in size and propagated east (Fig.1g).

The close succession of cyclones over the North Atlantic in the second part of February 2019 disrupted the normal progression of weather systems in the North Atlantic-European region. Both before the onset and during the blocking event, several rapidly intensifying cyclones and smaller low-pressure systems developed over the ocean (Fig. 2). Prior to the formation of the block, a rapidly intensifying cyclone (LE0, Fig.1a and blue line in 2a) led to the expansion of an upper-level ridge. This low-pressure system initiated the sequence of the cyclones that played an important role in the present case study. Moreover, it
triggered an intense CAO and strong surface evaporation events over the western North Atlantic (Fig.3a). The second rapidly intensifying cyclone developed on 18 February (LE1, Fig.1c, light blue line in Fig. 2a) over the Gulf Stream region and crossed the North Atlantic toward the southern tip of Greenland. The pressure drop during its intensification phase reached 39 hPa within 24h. Moreover, the divergent outflow caused by the WCB of this cyclone modified the upper-level NPVA (black crosses in Fig.1c), leading to its growth in size. In the lower troposphere, the passage of this cyclone resulted in a strong CAO event
over the Gulf Stream region (Fig.3b). The northward progression of the LE1 cyclone on 21 February was followed by the development of several secondary, much weaker cyclones in the central North Atlantic (L1, L2, Fig,2b). Despite their weakness, they acted to maintain the advection of cold air across the ocean (Fig.3b) and reinforce the upper-level NPVA (black crosses in Fig.1e) through cross-isentropic ascent.

The third rapidly intensifying cyclone emerged on 21 February (LE2, green line in Fig.1e, 2a) and followed a similar path as
the one from 18 February. The highest pressure drop observed for this system was 25 hPa within 24 h. The latent heat release in the ascending air stream of this cyclone led to the formation of a minor NPVA (Section 2.1.3, blue shading in Fig. 1e) that later joined the main NPVA (Fig. 1g) and reinforces the block. It is important to highlight the fact that cyclone LE2 propagated into the region of high surface fluxes left behind by cyclone LE1 (Fig.3b). Furthermore, the advection of cold air behind the cold front of LE2 resulted in another strong surface evaporation event over the Gulf Stream region (Fig.3c).

The last rapidly intensifying cyclone had genesis on 26 February (LE3, purple line in Fig.1g, 2a). The strengthened outflow associated with the cyclone's ascending air stream may have contributed to reinforcing the upper-level NPVA from western



direction (gray shading and black crosses in Fig.1g). Similar to the previously described low-pressure systems, it also caused a CAO event (Fig.1c). The extreme cyclones of 15, 18, 21, and 26 February were accompanied by weaker low-pressure systems developing in the central part of the North Atlantic (L1-L5, Fig.2b). Their development led to the expansion of CAOs further

into the ocean and the widespread occurrence of strong upward surface fluxes in the area (Fig.3).

This synoptic overview suggests that a series of factors contributed to the creation of the block in February 2019 and the resulting record-high temperatures. In the following sections, we will study in detail the importance of the described cyclones and CAO events for the development of the atmospheric block in February 2019.

## 3.2   Connection between the Gulf Stream region and the large scale dynamics

To investigate a potential link between the Gulf Stream region and the upper-level circulation during the blocking episode, we investigate the characteristics of the two main sets of trajectories (Section 2.2). Figure 4 shows properties of the air parcels that start from the upper-level NPVA objects (NPVA trajectories, Tab.1, Fig. 4a) and from the Gulf Stream (GS trajectories, Tab. 1, Fig. 4, b). Note that the times in Fig. 4 refer to the time when the trajectories are started in the NPVA objects and the Gulf Stream, respectively. As stated in Section 2.2, these trajectories ascend by at least 500 hPa prior to arriving in the

NPVA objects (for NPVA trajectories) and after starting from the GS region (for GS trajectories). Those base sets represent 53% of all trajectories started from NPVA objects (NPVA trajectories) and 43% of all trajectories started from the GS region (GS trajectories).

First, we determined the fraction of NPVA trajectories passing over the Gulf Stream in the lower troposphere (NPVA GS trajectories, Tab. 1) and of the GS trajectories ending up in the NPVAs in the upper troposphere (GS NPVA trajectories, Tab.

1). On average, more than 23% of the NPVA trajectories interact with the ABL over the Gulf Stream region (black, dashed line in Fig. 4a) and 61% of the GS trajectories travel into the NPVAs forming the block (black, dashed line in Fig. 3b). A more detailed analysis of the temporal evolution reveals that the fraction of the NPVA GS trajectories changes only by a maximum of $\pm$ 10% throughout the lifecycle of the block and does not fall below 18%, nor exceed 40% (Fig.4a). More variations are found in the fraction of the GS NPVA trajectories within all ascending GS trajectories (black, dashed line in Fig. 4b). 10 days

prior to the onset of the block, 30% of GS trajectories travel into the upper-level NPVA. In the following days, as we get closer to the blocking event, this fraction increases to 90% on 18 February. The increases in the fraction of trajectories ending up in the upper-level block are related to the development of cyclones, first from 13 February (LE0) and later the one forming on 18 (LE1, Fig.2a) and 21 of February (LE2, Fig.2a). Once we get closer to the end of the blocking event the number of the GS NPVA trajectories starts to decrease. This decline coincides with the shrinking of the upper-level NPVA and its eastward

propagation. On 28 February, only ∼20% of the GS NPVA trajectories ascend into the blocking region.

In the next step of our study, we focus in more detail on the airstream characteristics of the NPVA GS trajectories and the GS NPVA trajectories (Tab.1). This analysis is carried out by dividing the trajectories into different airstreams: CAO, WCB, DI, and DH (Section 2.2.5; Tab.1). As previously described in Section 3.1 the development of multiple cyclones over the Gulf Stream region throughout the second part of February 2019 provided conditions favorable for the advection of cold air

from higher latitudes across the warm waters of the western North Atlantic, particularly south of the Gulf Stream (Fig.4).



Trajectories passing through the Gulf Stream region often coincide with a CAO event, as shown by the high percentage of CAO trajectories in both NPVA GS and GS NPVA trajectories, which were approximately 67% and 89%, respectively. To gain further insight, we divided the CAO trajectories into subsets based on their strength, distinguishing between very weak (blue in Fig.3) and stronger (teal in Fig.4) characteristics (see Section 2.2.5). It's worth noting that the percentage values mentioned refer specifically to the strong CAO subsets. Furthermore, the fraction of CAO trajectories in both subsets remained high throughout the case study, indicating that CAO events were prevalent in the western North Atlantic in February 2019. CAOs lead to a large exchange of heat and moisture between the ocean and atmosphere, warming and moistening the air parcels. Consequently, almost all of the NPVA GS and GS NPVA trajectories that interacted with or were started from the Gulf Stream are diabatically heated by at least 2 K (orange in Fig.3). Steinfeld and Pfahl (2019) found in their 38-year global analysis of backward trajectories started in the upper-level blocks that the fraction of diabatically heated trajectories exceeds 30-45%, in contrast to a larger fraction obtained by Yamamoto et al. (2021) ∼51,8%, who attributed the higher values to the use of different blocking definitions and trajectory setup. When considering all NPVA trajectories released from the upper level NPVA objects after initial filtering (Section 2), we found, for this particular case, that ∼49% of them have experienced diabatic heating 3 days prior to the arrival in the blocking region. Moreover, when considering NPVA GS and NPVA nonGS datasets together, therefore after the application of the ascent criterion, this number increases to ∼63%. Such a large fraction of trajectories undergoing diabatic heating, similar to those obtained in both climatological studies, demonstrates the importance of diabatic processes and their potential role in the development of European Blocking in February 2019.

Next, we consider the trajectories that experience rapid ascent into the upper troposphere of 600 hPa within 48 h (green in Fig.4; Madonna et al., 2014). On average 24% of the GS and 30% of the NPVA trajectories are classified as WCB trajectories. A closer look at the temporal revolution of the WCB fractions reveals that for the NPVA trajectories, the percentage of WCB trajectories remains higher than 20% for the whole duration of the block and between 22-23 February it exceeds 40%, whereas temporal variations in the proportion of the GS WCB NPVA trajectories (green in Fig.3b; Tab.1) shows more fluctuation. Moreover, there appears to be a connection between the increases in the fraction of WCB trajectories and the growing number of trajectories defined as GS CAO NPVA (blue and teal in Fig.3b; Tab.1).

A significant portion of the NPVA GS and GS NPVA trajectories undergoes rapid descent of 400 hPa within 48 hours, satisfying the criterion for dry intrusions (violet in Fig.4; Raveh-Rubin, 2017). Dry intrusions are associated with exceptionally strong surface sensible and latent heat fluxes, resulting in moistening of air parcels before they ascend again into the upper troposphere. The examination of the spatial distribution of detected DIs, for both the NPVA GS and the GS NPVA trajectories, revealed that dry, cold air descending from the upper troposphere interacts with the boundary layer mostly in the confined region northeast of the Florida peninsula, over the southern section of the Gulf Stream. This region stands out in the analysis of moisture sources for the GS trajectories presented in the next section (Fig.6c, Section 3.3).

The fractions of each of the analyzed air streams, with an exception of DH, change slightly throughout the studied period (Fig. 4). Those temporal variations are less pronounced for the NPVA trajectories, compared to the GS ones. Overall, such fluctuations in the number of trajectories identified as CAO, WCB, or DI (Fig.4) are related to the formation and intensity of extratropical cyclones developing in the northwestern North Atlantic. The rapid intensification of cyclones (Fig.2a, LE0-





LE3) tends to be associated with higher fractions of WCB airstreams (Fig.4; Binder et al., 2016). For instance, the increases in the fraction of WCB trajectories around 23 February for the NPVA GS trajectories and 21 February for the GS NPVA trajectories are related to the rapidly intensifying cyclones from 18 (LE1) and 21 February (LE2, Fig.2a). The match between the fractions of the CAO and WCB trajectories (Fig.3b) found for the GS NPVA trajectories indicates that there is a potential

link between intense ocean evaporation events and the development of the WCB. Similar observations have already been reported in several studies (e.g. Aemisegger and Papritz, 2018; Pfahl et al., 2014; Eckhardt et al., 2004). The relatively high number of DI trajectories on 21 and 22 February for the NPVA GS trajectories and on 14 to 17 February for the GS NPVA trajectories indicates that those DIs might have provided conditions that favored the formation of the rapidly intensifying cyclones developing on 18 and 21 February (Fig.2a; Browning, 1997; Raveh-Rubin, 2017).

The previously described results suggest a strong connection between the CAO-induced air-sea interactions over the Gulf Stream and the large-scale dynamics associated with the formation of the block in February 2019. While it is widely established that diabatic heating in regions of intense surface heat fluxes influences the large-scale atmospheric circulation (e.g., Pfahl et al., 2015; Yamamoto et al., 2021; Tilinina et al., 2018), the understanding of the mechanistic link between processes that take place within the cold sectors of cyclones and upper-level ridge formation is still missing (Czaja et al., 2019). Our results show

that the intense, CAO-induced air-sea interactions in the western North Atlantic and an episode of European Blocking might be inherently linked. Furthermore, the connection between the surface fluxes and coherent air streams hints at a dynamical linkage of the Gulf Stream front to the large-scale atmospheric circulation. In the following section, we aim to further detail this mechanistic link.

### 3.3  Moisture sources for NPVA GS trajectories

The rapid, cross-isentropic ascent of air parcels into the upper-level NPVA is driven by the latent heat release during cloud formation and precipitation (Joos and Wernli, 2012). For clouds and precipitation to form, a sufficient moisture supply is needed (Eckhardt et al., 2004; Pfahl et al., 2014). Using the method of (Sodemann et al., 2008) we analyzed the sources of moisture for the ascent of the NPVA GS trajectories, and for those that did not directly interact within the Gulf Stream region in the lower troposphere -labeled as NPVA nonGS trajectories (Tab.1, Section 3.4).

We will first focus on the NPVA GS trajectories (see Table 1). On average, these trajectories gain moisture about 3.5 days prior to their ascent, as indicated by the orange bars in Figure 5a and Table 2. Approximately 78% of moisture uptakes take place within the first seven days prior to ascent, with the largest contributions found within the first five days (about 60%, indicated by the orange line in Figure 5a and Table 2). For many trajectories, moisture is supplied shortly (24 hours) prior to ascent, as shown by the orange bars in Figure 5a. Over the course of 10 days, 75% of the uptakes take place over the ocean and

25% over land, as detailed in Table 2. On average, NPVA GS trajectories take 2.6 days from the start of ascent to arrive in the upper-level NPVA, as indicated by the orange bars in Figure 5b and Table 2. These results indicate that moisture uptake for air parcels passing over the Gulf Stream, subsequent ascent, and arrival in the upper-level NPVA occur in rapid succession

The spatial distribution of moisture uptakes for the NPVA GS trajectories is presented in Figure 6c. The highest frequency of moisture uptakes for NPVA GS trajectories is found in the region of the Gulf Stream and in the central North Atlantic. Figure



6a and b indicates that moistening and heating of air masses in this stretch of the ocean are related to the air-sea interactions taking place during CAOs - triggered by the passing, rapidly intensifying cyclones (Fig.2a; Fig.3).

The method for moisture source identification used in our analysis enables the calculation of moisture uptakes contribution from each source area (Fig.6e). It is noteworthy that, despite the backward extension of trajectories used to identify moisture sources, the majority of uptakes occurred quite close to the region of the block. Moreover, moisture sources directly from the

area of the Gulf Stream accounted for 80-90% of the uptakes, together with the uptakes taking place in the eastern part of the Gulf of Mexico. The uptakes in the central North Atlantic contributed about 20% to total moisture content prior to ascent. Overall, the primary contributions can therefore be attributed to the area of the strongest CAOs (Fig.6a), aligned with the Gulf Stream SST front.

Overall, CAOs play a crucial role in the life cycle of NPVA GS trajectories. On average, 56% of all moisture uptakes occur

during CAO events over the ocean when the atmosphere is colder than the surface (dark blue line in Fig. 7a). Around 16% of these CAOs overlap with the uptake location in the cold sector of the cyclone (green line in Fig. 7a). Moreover, a substantial proportion of moisture uptakes transpire during more intense CAOs (2K, light blue, dashed line in Fig. 7a), accounting for over 31% of all uptakes. Only about 5% of the uptakes occur in the cold sector of the cyclone when it is not associated with a CAO (depicted by the brown line in Fig. 7a). Figures 7 a and b demonstrate that a high fraction of CAO uptakes coincide with

upward SLHFs exceeding 100 W/m$^2$.

Moreover, these high heat flux events predominantly occur either before or during the formation of rapidly intensifying cyclones, as shown in Figure 2a. For example, the proportion of CAOs and the intensity of SLHFs were remarkably high between February 12 and 15, preceding the rapid intensification of Cyclone LE0. Another similar event was observed around February 21, just before the rapid growth of LE2. The most recent peak in the CAO fraction, accompanied by intense surface

evaporation, took place before the intensification of cyclone LE3 around February 25-26 (Figures 2a and 7). The relatively small number of uptakes occurring in the cold sector of the cyclone (brown and green lines in Figure 7a) compared to the large fraction of uptakes in CAO regions implies that uptakes primarily occur in the CAO induced by the cyclone, rather than directly in the cold sector of the cyclone causing the CAO. It is worth noting that the episodes of extreme SLHF are significantly larger in magnitude for the NPVA GS trajectories than for the NPVA nonGS trajectories (Figure 7b).

Our results are in agreement with other studies (e.g. Papritz and Grams, 2018; Aemisegger and Papritz, 2018; Hawcroft et al., 2012), indicating that CAOs play an important role in the water cycle of cyclones. However, it remains poorly understood how air parcels moistened in the region behind a passing cyclone's cold front end up in the upper-level NPVA. One possible explanation is the existence of the so-called 'hand-over' mechanism described in detail by Papritz et al. (2021). They found that moisture precipitating in deep North Atlantic cyclones originates in the cold sector of a preceding cyclone and is fed into

the ascent regions of the subsequent cyclone via the feeder air stream (Dacre et al., 2015).

To better understand moisture transport mechanisms in our case study, and explore the possibility of the 'hand-over' mechanism, we analyzed the relationship between surface latent heat flux (SLHF) and atmospheric trajectories. Following the methods of Yamamoto et al. (2021) and Tilinina et al. (2018), we identified regions of maximum SLHF experienced by trajectories





to locate areas of most intense evaporation and heating. We also examined the locations where the trajectories ascend into the upper troposphere begins (Fig.8).

Figure 8a (red contours) illustrates the analysis for 24 February at 21:00 UTC, revealing that the most intense evaporation occurs near the Gulf Stream and areas with the highest CAO index values. Trajectories do not ascend immediately but remain in the atmospheric boundary layer for at least 24 hours, being advected south and southeast with the cold air in the cyclone's cold sector (green contours in Fig.6a). The ascent occurs approximately 54 hours after the maximum SLHF values (blue contours in Fig.6b), suggesting that the ascent is not directly caused by the cyclone responsible for strong surface evaporation. Instead, our findings suggest that cyclone LE2 (Fig.2a) and secondary cyclones L1 and L2 (Fig.2b) play a significant role in lifting the moistened air parcels into the upper troposphere's NPVA, which is consistent with the results reported by Papritz et al. (2021). Cyclone LE2 travels through and intensifies in the region of strong CAO left behind by LE1 (Fig.3b). The already moistened air is then fed into the ascending airstream of the LE2 cyclone. Additionally, cyclones L1 and L2 develop behind LE1 lifting some of the air masses into the upper troposphere.

To confirm that the process described above dominates in the present case study, we performed the analysis shown in Figure 8a for all the NPVA GS trajectories (Fig.8c). Analyzed trajectories experience the most intense moistening along the Gulf Stream SST front. One day later, the moistened air moves south or southeast, together with the air in the cyclone's cold sector. Trajectories begin their ascent into the upper troposphere on average 3.5 days after reaching maximum SLHF values (Tab.2). The time difference between the maximum SLHF and the time of ascent is consistent with a hand over from one cyclone to another. In line with the findings of Papritz et al. (2021), air parcels collect moisture in the region of intense surface evaporation behind the cold front of a cyclone. Their ascent into the upper troposphere is possible thanks to the passage of a second cyclone, that travels into the region of high surface fluxes. This notion is further supported by the observation that more than 76% of the NPVA GS trajectories start their ascent more than a day after passing through the region of the most intense SLHF. Furthermore, they remain in the ABL for ∼4 days prior to the ascent and are continuously subjected to surface evaporation for ∼2.5 days (Tab.2). Such a long exposition to negative SLHF supports the notion that they are advected with the cold air behind the cyclone's cold front. Given those statistics, as well as the dominance of the CAO trajectories in the NPVA GS dataset (Section 2.2.5), we propose that during the study period, the 'hand over' process is the dominating pathway for moisture transfer in the North Atlantic for air masses that interact directly with the Gulf Stream.

## 3.4 Moisture sources for NPVA nonGS trajectories

For backward trajectories that did not interact with the boundary layer over the Gulf Stream, the primary moisture sources are located in the Gulf of Mexico and the Caribbean Sea (Fig.8d), with smaller contributions found in the North Atlantic and the eastern Pacific Ocean.

Similar to NPVA GS trajectories, NPVA nonGS trajectories acquire moisture around 3.8 days prior to ascent (shown as blue dashed bars in Fig.5a and Tab/2). The majority (67%) of this moisture is acquired within the first seven days of ascent, with 48% being acquired within the first five days (blue line in Figure 5a). NPVA nonGS trajectories tend to have a higher number of uptakes compared to NPVA GS trajectories, accounting for 75% of all NPVA trajectories. However, only 44% of these uptakes





|  | NPVA GS trajectories | NPVA nonGS trajectories |
|---|---|---|
| Average time of moisture uptake prior to the start of ascent. | 3.5 days | 3.8 days |
| Fraction of moisture supplied within first 5 days. | 60% | 48% |
| Fraction of moisture uptakes over the ocean. | 78% | 44% |
| Average time of ascent start in relative the time of arrival into the NPVA. | -2.6 days | -6.65 days |
| Average time in the ABL prior to ascent. | 4 days | 3.8 days |
| Average time of continuous SLHF<0 W/m$^2$ prior to the start of ascent. | 2.5 days | 23 h |

**Table 2.** Summary of general characteristics and moisture sources for NPVA GS and NPVA nonGS trajectories.

occur over the ocean, which is significantly lower than the 75% of trajectories that interact with the Gulf Stream (Tab.2. NPVA nonGS trajectories take an average of 6.65 days to ascend into the upper-level NPVA from the time they leave the atmospheric
boundary layer (shown as blue dashed bars in Fig.5b). This suggests that the area of ascent for these trajectories differs from that of the NPVA nonGS trajectories (as shown in Fig.8c).

The CAO index (Fig.6a) shows that weak CAOs in February 2019 took place also over the eastern Pacific Ocean. In consequence, on average 45% of NPVA nonGS trajectories experienced a CAO. Around 13% of those moisture sources were located over the ocean, in the cold sector of the cyclone. Surface evaporation events were not as intense or as frequent for the NPVA
nonGS trajectories as they were for the NPVA GS trajectories (blue colors in Fig.7b). This is also confirmed by the spatial distribution of CAOs (Fig.6a), SLHF (Fig.6b), and moisture sources (Fig.6d). The regions supplying moisture for the NPVA nonGS trajectories are identified as areas of weak or nonexistent CAOs and substantially weaker surface fluxes (Fig.6a,b). The highest moisture contributions are found in the subtropical regions of the Gulf of Mexico and the Caribbean Sea (Fig.6f). It is noteworthy that, in contrast to the uptakes for NPVA GS trajectories, the uptakes in the Gulf of Mexico are located further west
(Fig.6e-f). Moistened air masses of NPVA nonGS trajectories remain in the atmospheric boundary layer for ∼3.8 days prior to ascent, but maintain negative SLHF for only ∼23 h (Tab. 2)

The NPVA nonGS trajectories have two primary regions of ascent (Fig.8, d). Air masses that experience intense moistening in the subtropical Caribbean Seas start their ascent into the upper troposphere over the southern coast of the United States (Fig.8, d). Whereas the trajectories with the highest SLHF over the eastern Pacific begin their upward movement in the same
area where they pick up moisture (Fig.8d; Fig.6d). In comparison to the NPVA GS trajectories, which ascend straight into the upper-level NPVA, the locations of the ascent of NPVA nonGS trajectories are further away from the blocking region. This explains why the time needed for the NPVA nonGS trajectories to reach upper-level NPVA, from the time they leave the atmospheric boundary layer, is much longer (∼6.65 days). Upon ascending into the upper troposphere, those air masses are then advected into the block and do not interact directly with the extratropical cyclones developing in the North Atlantic or
CAOs in the Gulf Stream region.




## 3.5 Properties of GS NPVA trajectories

The results shown above indicate that the water cycle of extratropical cyclones in the North Atlantic is substantially influenced by the air-sea interactions over the Gulf Stream. The significance of the processes taking place over the warm waters of the Gulf Stream can be established even more clearly by taking a closer look at the temporal variations in the GS NPVA trajectories'

properties. In this section, we will focus on the GS NPVA trajectories, since their higher spatial resolution (50 km) is more suitable for an in-depth analysis of the properties of air that have passed over the Gulf Stream and contributed to the formation of the block over western Europe.

Analysis of pressure changes along the trajectories shows that ascent towards the block takes place within 2-3 days after leaving the Gulf Stream region (Fig.9a). Air masses that interact with the warm waters of the Gulf Stream typically originate

from the mid and lower troposphere, and upon passing through the western North Atlantic, travel into the NPVAs in the upper atmosphere. The potential temperature changes during the ascent highlight the importance of diabatic processes, which, as previously mentioned, occur in all of the GS NPVA trajectories (Fig.9b, Fig.4b).

Furthermore, the high sensible and latent heat fluxes experienced by the trajectories as they interact with the atmospheric boundary layer over the Gulf Stream can be attributed to the influence of CAOs (Fig.9c-d). The advection of cold air across

the warm waters of the Gulf Stream leads to significant heat and moisture exchange between the ocean and the atmosphere, resulting in an increase of specific humidity (Fig.8f). The period of strong moistening and heating of GS NPVA trajectories in the lower troposphere coincides with a significant decrease of potential vorticity (Fig.8e). In fact, we determined, that a large number of trajectories have negative PV in the lower troposphere. It is important to highlight that negative PV anomalies only occur for a limited number of time steps and at different time steps for each of the considered trajectories. Consequently, when

visualizing the median, 10th, and 90th percentiles in plots (such as Fig.9), these negative PV anomalies may not be evident.

Nevertheless, the presence of negative PV suggests that very intense mixing or cloud-related evaporative processes are taking place in the cold sector of cyclone (Attinger et al., 2019; Crezee et al., 2017; Attinger et al., 2021). Furthermore, low potential vorticity air in the inflow region of the WCB can significantly impact the development of NPVAs in the upper troposphere (Methven, 2015; Teubler and Riemer, 2016). Therefore, a closer examination of the changes in PV along GS NPVA trajectories

is carried out.

### 3.5.1   Negative Potential Vorticity in the lower troposphere

Overall, approximately 76% of the GS NPVA trajectories have negative PV in the lower troposphere over the ocean at some time prior to ascent. In agreement with these results, when the same analysis is applied to the NPVA GS trajectories - 82% have negative PV in the atmospheric boundary layer.

To explore the role of air parcels with negative PV in the formation of upper-level negative PV anomalies (NPVAs) we divided the GS NPVA trajectories into two subsets: those with negative PV (GS NPVA negPV trajectories, 76% of all GS NPVA trajectories) and those with continuous positive PV (GS NPVA posPV trajectories, 24% of GS NPVA trajectories) in the lower troposphere. To investigate the potential influence of negative PV in the lower troposphere on the formation of



upper-level NPVA, we examined the inflow and outflow stages of ascent. Specifically, we re-centered the time evolution of the
trajectories on the level of maximum heating, which is indicative of the release of latent heat during upward air mass movement.
By comparing the two sets of trajectories, we aim to determine whether the presence of negative PV air in the inflow stage of
the ascending air stream leads to the formation of low PV air in the upper troposphere. Our results show that the GS NPVA
negPV trajectories are located in the lower layers of the troposphere (Fig.10a) and experience more intensive heating during the
ascent (Fig.10b). Without indicating a cause-and-effect connection, greater fluxes in the inflow stage (Fig.10c-d) and elevated
moisture content during the ascent (Fig.10f) co-occur with a rise in heating intensity throughout the ascent. Interestingly,
despite experiencing negative values of PV in the atmospheric boundary layer and a strong heating rate, the PV of the GS
NPVA negPV trajectories is not lower than that of the GS NPVA posPV trajectories in the upper troposphere (Fig.10e) nor do
they reach a higher outflow height (Fig. **??**a,b). In fact, the GS NPVA negPV trajectories typically begin at a lower altitude,
and as a result, more heating is required for these trajectories to achieve a comparable outflow height to that of the GC NPVA
negPV trajectories. Surprisingly, the PV values in the GS NPVA negPV trajectories are even slightly higher when reaching
the upper troposphere. Additionally, we observe that the air masses maintaining positive PV values interact with the CAOs
to a lesser degree, as evident from the temporal changes of sensible heat flux in the two types of trajectories. Indicating that
processes occurring during CAOs may be responsible for the decrease of PV in the atmospheric boundary layer.

There are several processes that can result in the destruction of PV in the lower troposphere, including friction, evaporative
cooling, sublimation of snow, snow melting, or turbulent fluxes (Crezee et al., 2017; Attinger et al., 2019, 2021). To establish
what mechanism leads to the PV destruction throughout our case study, we examined vertical cross sections of cloud liquid
water content and potential vorticity over the area of the Gulf Stream. For the purpose of this analysis, we used the ERA5
reanalysis dataset with a higher temporal resolution of 1 hour. Obtained results reveal that the air parcels with negative PV
in the lower troposphere are primarily located below liquid water clouds (Fig.11b), in the cold sectors of the cyclones (Fig.2,
Fig.11a). The cold sector can be recognized in Figure 11b at approximately 36°N latitude by looking at the cloud structure.
Clouds in the warm sector of the cyclone extend deep into the atmospheric boundary layer, while the cold sector is dominated
by low-level stratiform clouds. Low-level clouds forming during the advection of cold air over the ocean due to the cooling
of the surface are classified as stratiform clouds (Painemal et al., 2021). The presence of air parcels with negative PV in those
areas suggests that evaporative cooling is the main cause of PV reduction. This was confirmed by Chagnon et al. (2013), who
discovered that evaporative cooling in the air descending behind the cold front decreases PV. This idea is further reinforced by
the studies of e.g.,Wood (2005), Jensen et al. (2000) and Paluch and Lenschow (1991), who found that evaporative cooling in
the sub-cloud layer of stratiform clouds is often triggered by the cooling that results from drizzle evaporation.

In most of the analyzed timesteps (e.g. Fig.10) the PV in the lower troposphere does not exceed -1 PVU. However, for
several air parcels, we found values below -2 PVU in the two lowest model layers right behind the cold front. Attinger et al.
(2019) and Vannière et al. (2017) attribute the prevalence of negative PV along the cold front to unstable conditions and high
surface fluxes. This likely also applies to our case, as the high negative PV values are found at low altitudes, in the regions
of very intense surface fluxes, mainly during the intensification stage of the extreme cyclones (Fig.2a). Overall, we presume





that the combination of strong surface fluxes, heating from the surface and evaporative cooling from low clouds leads to the development of a highly unstable environment, making the presence of negative PV in our case so widespread.

It is worth highlighting that a significant number of air parcels in Figure 11a have negative PV and are positioned ahead of the cold front. Our analysis of consecutive time steps reveals that these parcels are transported to this location due to the advection of cold air that trails the cold front from a preceding cyclone. Considering the handover mechanism's predominance in our case study and the findings presented in Figure 10, it is reasonable to expect that their PV will increase within the next few hours and they will be carried upwards into the upper troposphere by the ascending airstream of cyclone LE1 (Fig. 2).

## 4    Synthesis and Discussion

Our study suggests that air-sea interactions over the Gulf Stream are closely linked to the development of the European Blocking event in February 2019. Furthermore, our findings raise the possibility that a similar relationship may exist at a climatological level, indicating that air-sea interactions over the Gulf Stream could be relevant for other blocking events in the North Atlantic-European region. Trajectory analysis reveals that on average 60% of all GS trajectories started from the Gulf Stream

end up in the blocking region, representing approximately 23% of all trajectories that formed the NPVAs. The potential importance of those air masses for the development of the block can be established based on the results of Steinfeld et al. (2020), who determined that critical features of the block, including extent, strength, and lifetime, are strongly affected by latent heating taking place in the ascending airstreams. Our analysis revealed that all of the Gulf Stream trajectories (representing ∼23% of all trajectories) experience diabatic heating during the first three days after starting from the blocking region. Given that approx-

imately 49% of trajectories released from the NPVAs undergo diabatic heating, our results suggest that air masses interacting with the Gulf Stream may play a considerable role in the blocking formation (see Section 2.2.5).

Our analysis shows that air moistened and warmed in the CAOs generated by the passage of a cyclone is lifted into the upper troposphere with the ascending air stream of a consecutive cyclone, in agreement with Papritz et al. (2021) and Sodemann and Stohl (2013). This mechanism is schematically depicted in Fig. 12. The passage of a cyclone marked as LE1 (Fig.2a)

over the western North Atlantic results in the development of a CAO behind the cold front (Fig.12a), providing conditions for the moistening and heating of the atmospheric boundary layer. The strongest fluxes are found along the Gulf Stream SST front, as the air-sea temperature contrast is the largest in this region. 30 hours later, cyclone LE1 is located near the southern tip of Greenland, while two smaller cyclones L1 and L2 developed in the central North Atlantic (Fig.12b). The ascending air streams of those cyclones lift some of the air masses, that were moistened and heated in the cold sector of cyclone LE1, into

the upper troposphere. Moreover, the air-sea interactions in the cold sectors of cyclones L1 and L2 result in further heating and moistening of the atmospheric boundary layer in the central North Atlantic. At the same time, a third cyclone develops, LE2, which travels into the region of large surface latent and sensible heat fluxes created by cyclones L1, L2, and LE1. The air, moistened thanks to the advection of cold air by these cyclones is fed into the warm sector of cyclone LE2 and begins its cross-isentropic ascent into the upper troposphere. Therefore, in agreement with Papritz et al. (2021) and Dacre et al. (2019), this

suggests that one cyclone provides the environment favorable for the intensification of a consecutive cyclone. The air masses,



heated and moistened in the cold sector of the first cyclone, are later fed into the warm sector of the consecutive cyclone and eventually lifted into the upper-level NPVA.

The Gulf Stream region serves as an important moisture source for those NPVA trajectories that passed over it in the lower troposphere, in agreement with the results of Yamamoto et al. (2021) and Pfahl et al. (2014). In fact, the Gulf Stream
contributes most of the moisture present in the air prior to its ascent. Those findings imply that the moisture sources for extratropical cyclones in the North Atlantic have a regional character and are concentrated in areas where there is a strong ocean-atmosphere temperature contrast. Another area of moistening of the atmospheric boundary layer is found in the central North Atlantic, south of the Gulf Stream's eastward extension. This area is frequently affected by the advection of cold air from the Labrador Sea or the passage of a cyclone, which provides conditions for strong air-sea interactions. In fact, the
cyclones recognized as rapidly intensifying propagate into this stretch of the ocean, while several of the secondary cyclones originate there. The frequent presence of cyclones in a region of strong surface evaporation highlights the importance of 'preconditioning' for cyclone development described by (Papritz et al., 2021).

It should be noted that the subtropical region of the Caribbean Seas and eastern North Pacific serves as a significant source of moisture for air streams ascending to the block. However, it is worth emphasizing that the regions in which these air masses
ascend, namely the Gulf of Mexico and the Pacific Ocean, are situated outside the western North Atlantic. Consequently, remote sources of moisture do not appear to be relevant for the air masses ascending into the block within the extratropical cyclones that formed in the North Atlantic in February 2019.

The occurrence of CAOs during the studied period is primarily due to the advection of cold air behind cold fronts of passing cyclones that develop in the Gulf Stream region and its extension. These CAOs not only lead to intense surface heat fluxes,
which are essential for maintaining baroclinicity (Papritz and Spengler, 2015), but also contribute to the self-maintenance of the storm track (Aemisegger and Papritz, 2018) and the formation of rapidly intensifying low-pressure systems crucial for blocking growth (Colucci, 1985; Colucci and Alberta, 1996). Exceptionally strong oceanic evaporation events are vital for the development of WCBs (Pfahl et al., 2014; Eckhardt et al., 2004), which facilitate the transport of low PV air into the upper troposphere (Pfahl et al., 2015; Steinfeld and Pfahl, 2019; Methven, 2015).

According to a study by Papritz and Grams (2018), weather regimes modulate the occurrence of CAOs, but our research indicates that this relationship may be mutual. CAOs that develop shortly before and during blocking create favorable conditions for strong evaporation events in the western North Atlantic, providing the necessary moisture for the rapid intensification of cyclones and the development of secondary low-pressure systems. These cyclones contribute to the northward expansion of an upper-level ridge, weakening the zonal flow and fostering further CAOs development, which in turn trigger intense surface
evaporation events (Kautz et al., 2022; Gao et al., 2015).

The advection of cold air over warmer sea surfaces leads to air-sea interactions, which can result in the destruction of PV in the lower troposphere. Our research aligns with the findings of previous studies conducted by Crezee et al. (2017), Chagnon et al. (2013), and Attinger et al. (2019), which suggest that the presence of negative PV in the atmospheric boundary layer can be attributed to evaporative cooling beneath low-level stratiform clouds and strong surface fluxes.



In contrast to the hypothesis of Methven (2015), which proposes that the average PV of WCB outflow is nearly equal to the PV of its inflow due to an almost negligible net change in models, our study offers a slightly different viewpoint. In the analyzed case study, trajectories with negative PV in the atmospheric boundary layer (ABL) exhibit a somewhat higher PV in the upper troposphere when compared to trajectories with positive PV in the ABL. These findings imply that diabatic PV production and destruction may often not exactly balance during ascent as suggested by **?**. This highlights the potential need for further research on the relationship between diabatic processes and changes in ascending air streams.

Our study demonstrates that NPVA objects forming the block are partially created and maintained by ascending air streams of subsequent cyclones developing in the North Atlantic, as indicated by low values of PV in the upper troposphere (Figs.10 and 11). However, we cannot directly link the growth of NPVAs to the presence of negative PV in the atmospheric boundary layer. In our case study, negative PV functions more as a marker for an unstable environment and evaporative cooling associated with low-level stratiform clouds.

## 5 Conclusions

To summarize, our study provides a possible explanation for a mechanistic link between air-sea interactions over the Gulf Stream region and the formation of blocks over the North Atlantic and European regions. It highlights the importance of CAOs and associated moisture uptakes and subsequent ascending air streams for the formation of a quasi-stationary, upper-level ridge. It is clear that a single case study cannot be used to draw any general conclusions. However, considering the fact that singular aspects of our analysis are in agreement with recent publications focusing on moisture transport in the North Atlantic and the formation of blocks (e.g. Papritz et al., 2021; Aemisegger and Papritz, 2018; Hirata et al., 2019; Yamamoto et al., 2021; Steinfeld et al., 2020; Dacre et al., 2019), it provides a basis for further research. Additionally, an increasing number of recent studies indicate that biases in the SST representation in the North Atlantic are related to models' inability to correctly predict blocks (e.g. Athanasiadis et al., 2022; Czaja et al., 2019; Kwon et al., 2020), thus suggesting that the mechanistic linkage between the Gulf Stream, diabatic processes, and the large-scale extratropical circulation found in our results are relevant on a climatological scale. Therefore, in a subsequent study, we are going to analyze those relationships using a similar trajectory dataset spanning 40 years of ERA5 data. Using the methods applied in this case study, we will aim to establish whether the air-sea interactions over the Gulf Stream modulate the large-scale dynamics and formation of all seven weather regimes (Grams et al., 2017) and to identify the predominant way by which the signal from the lower troposphere is transferred to the upper-level flow.

*Author contributions.* MW, CMG, LP, MF planned and designed the case study. MW analyzed the data and wrote the manuscript. CMG, LP, MF gave important guidance during the project and provided feedback on the manuscript.



*Competing interests.* CMG and LP are members of the editorial board of Weather and Climate Dynamics. The authors have no other competing interests to declare.

*Acknowledgements.* We gratefully acknowledge the European Centre for Medium-Range Weather Forecasts (ECMWF) for providing the ERA5 reanalysis data set, which was made available through their website http://www.ecmwf.int. We also thank the members of the Large-Scale Dynamics and Predictability group at KIT, as well as Jamie Mathews and Arnaud Czaja from Imperial College London, for their valuable discussions and contributions to this project. Additionally, we extend our thanks to Heini Wernli and Michael Sprenger for providing the Lagranto toolkit and cyclone dataset, which greatly facilitated our analyses. The contributions of MW and MF are funded by the German Research Foundation (DFG; Grant GR 5540/2-1) and the Swiss National Science Foundation (SNSF; Grant 200021E_196978), respectively, as part of the Swiss-German collaborative project "The role of coherent air streams in shaping the Gulf stream's impact on the large-scale extratropical circulation (GULFimpact)." The contribution of CMG is funded by the Helmholtz Association as part of the Young Investigator Group "Sub-seasonal Predictability: Understanding the Role of Diabatic Outflow" (SPREADOUT, grant VH-NG-1243).



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



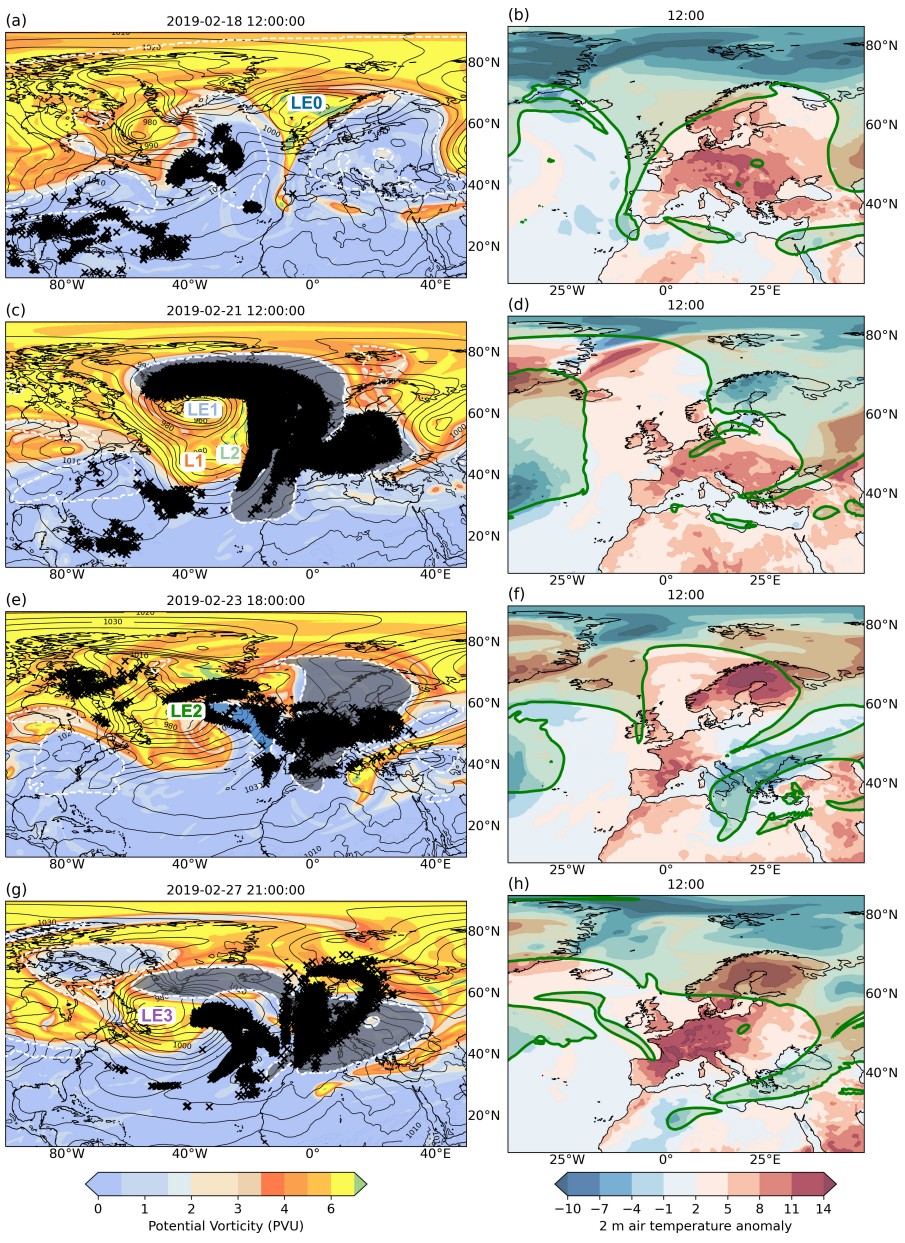

**Figure 1.** Synoptic evolution of European Blocking episode from February 2019. First column: potential vorticity (shading, PVU) at 315 K and negative potential vorticity anomaly objects (dashed contour) on 2019-02-18, 12:00 UTC (a), 2019-02-21 12:00 UTC (c), 2019-02-23 18:00 UTC (e), 2019-02-27 21:00 UTC (g). The major NPVA is shaded in grey and the minor in light blue (Section 2.1.3). Black crosses: locations of NPVA GS trajectories on the specified dates on the 315 K ($\pm$2K) isentropic level. Second column: 2 m temperature anomalies (with respect to a 30d running mean, shading) and upper level 2 PVU contour at 315K (green line), with PV values higher than 2 PVU shaded in green, for 12:00 UTC of each day from figures (a, c, e, g). LE0-3 and L1-2 refer to cyclones from Fig.2.



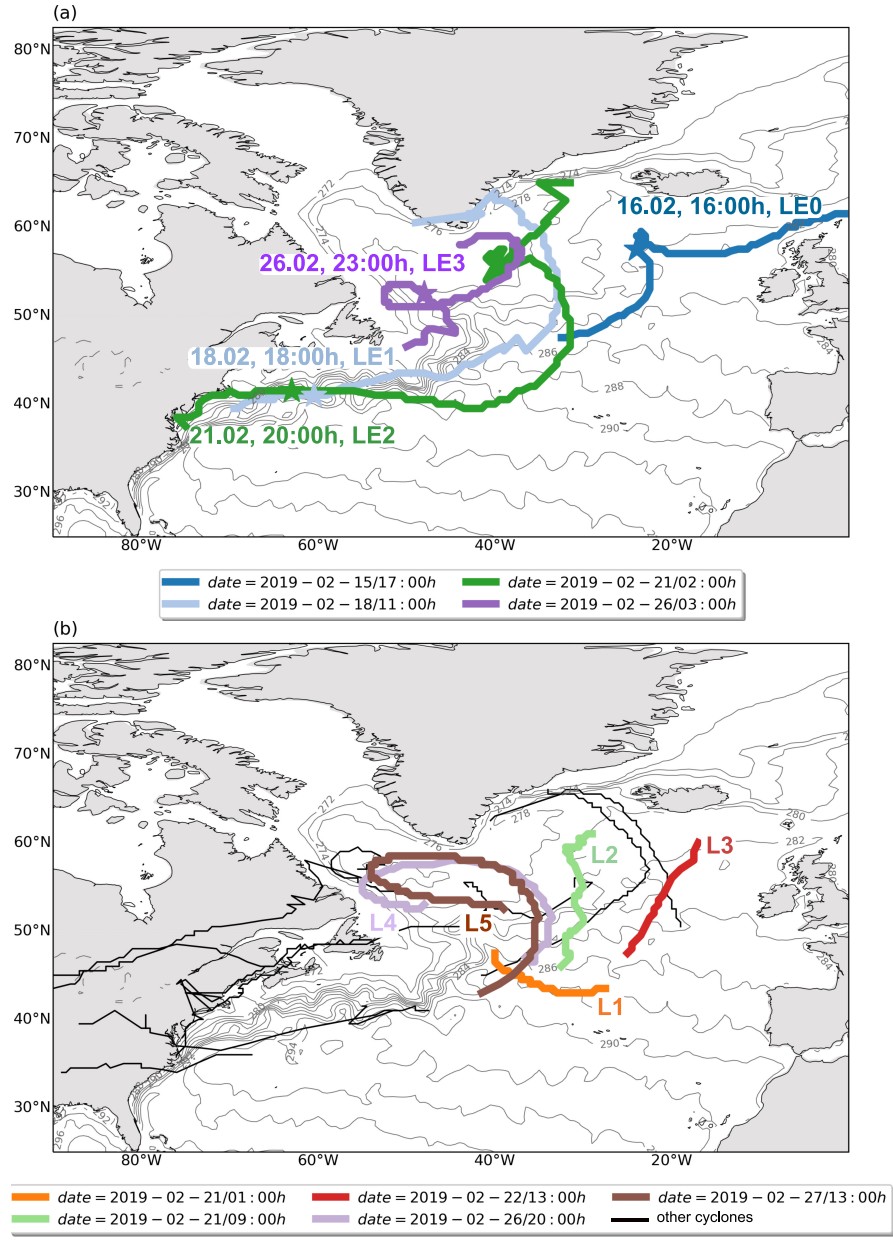

**Figure 2.** Tracks of cyclones formed in the North Atlantic between 15th and 27th Feb. 2019 together with contours of mean sea surface temperature throughout that period. (a) Rapidly intensifying cyclones (Sanders and Gyakum, 1980), the dates near the name of the cyclones e.g. 21,20:00, LE2 refer to the date and hours when the cyclones started to intensify rapidly; E in the name of the cyclone stands for extreme, (b) other cyclones, in color those relevant for the paper, in black other, non rapidly intensifying cyclones. The dates in the legend for both (a) and (b) indicate the start of the cyclone track.



**Figure 3.** Cold air outbreak index (blue shading) and mean sea level pressure (black contours) in the North Atlantic and eastern Pacific on the dates when cyclones LE1, LE2, LE3 (Fig.2) start to intensify rapidly: (a) 2019-02-18, 18:00, (b) 2019-02-21, 20:00, (c) 2019-02-26, 03:00. LE1, LE2, LE3, L1, L2, L4 indicate positions of cyclones from Fig.2 on the chosen date.

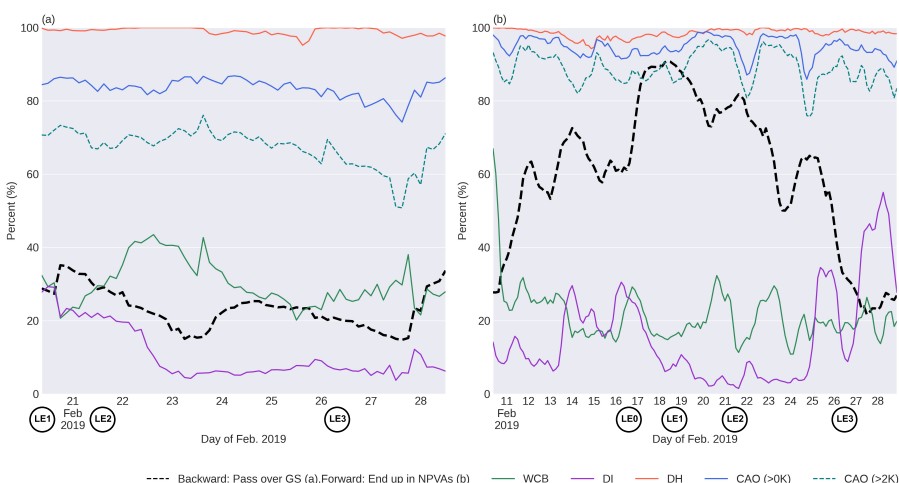

**Figure 4.** The percentage of different subsets of trajectories from Table 1. The horizontal axis represents the starting times for NPVA and GS trajectories. The black dashed lines indicate the fractions of NPVA GS and GS NPVA trajectories within NPVA and GS trajectories respectively. The colored lines represent different airstreams within the NPVA GS and GS NPVA trajectories. The green lines indicate NPVA WCB GS and GS WCB NPVA, the orange lines indicate NPVA DH GS and GS DH NPVA, the blue lines indicate NPVA CAO1 GS and GS CAO1 NPVA, the teal lines indicate NPVA CAO2 GS and GS CAO2 NPVA, and the purple lines indicate NPVA DI GS and GS DI NPVA. LE0-3 refers to the starting dates of cyclones in Figure 2.



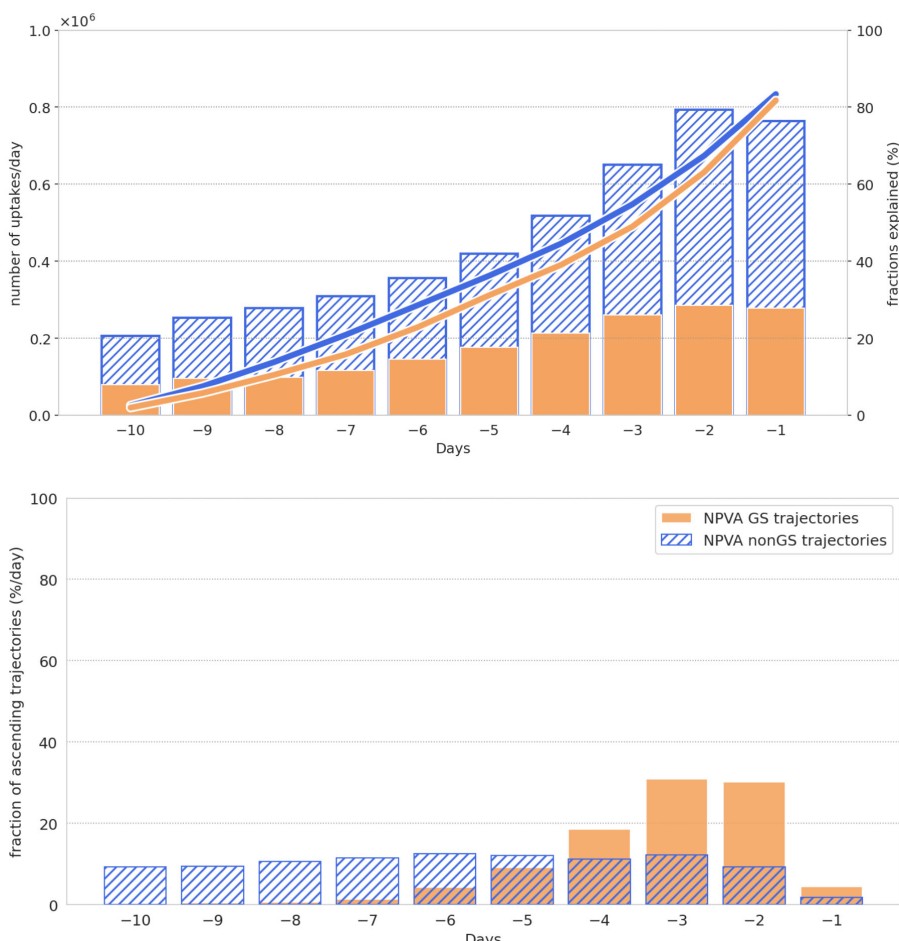

**Figure 5.** Characteristics of NPVA GS and NPVA nonGS trajectories. (a) Contributions of accumulated moisture uptakes to moisture present at the start of ascent (day 0) for NPVA GS (orange line) and NPVA nonGS (blue line) trajectories. Orange bars show the number of uptakes per day for NPVA GS trajectories and dashed, blue bars for NPVA nonGS trajectories.(b) Fraction of trajectories starting their ascent on a specific day, where day 0 represents the moment when they reach the upper level NPVA.



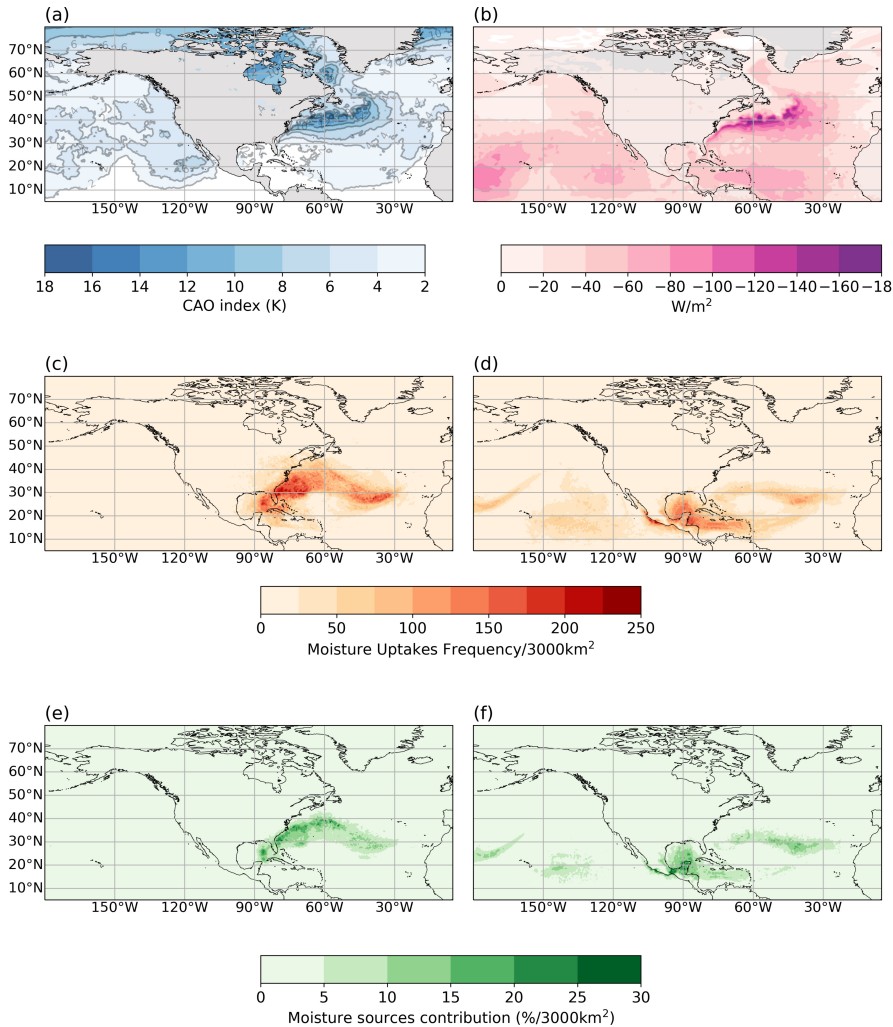

**Figure 6.** Analysis of moisture sources for two different subsets of NPVA trajectories (Tab.1). (a) Mean composite of CAO index ($\theta_{SST} - \theta_{850}$) for the period of 15 to 28 February (shading and contours), contours are plotted every 2 degrees from 2 to 20 K. (b) Mean composite of surface latent heat flux for the period of 15 to 28 February (shading), contours are plotted every -20 W/m$^2$. Negative SLHF in the ERA5 dataset indicates that SLHF is from the ocean to the atmosphere. Sources of moisture for the NPVA GS (left column) and NPVA nonGS trajectories (right column). (c-d) The frequency of moisture uptakes per 3000 km$^2$, (e-d) Moisture sources contribution to total moisture content prior to ascent (%3000 km$^2$).

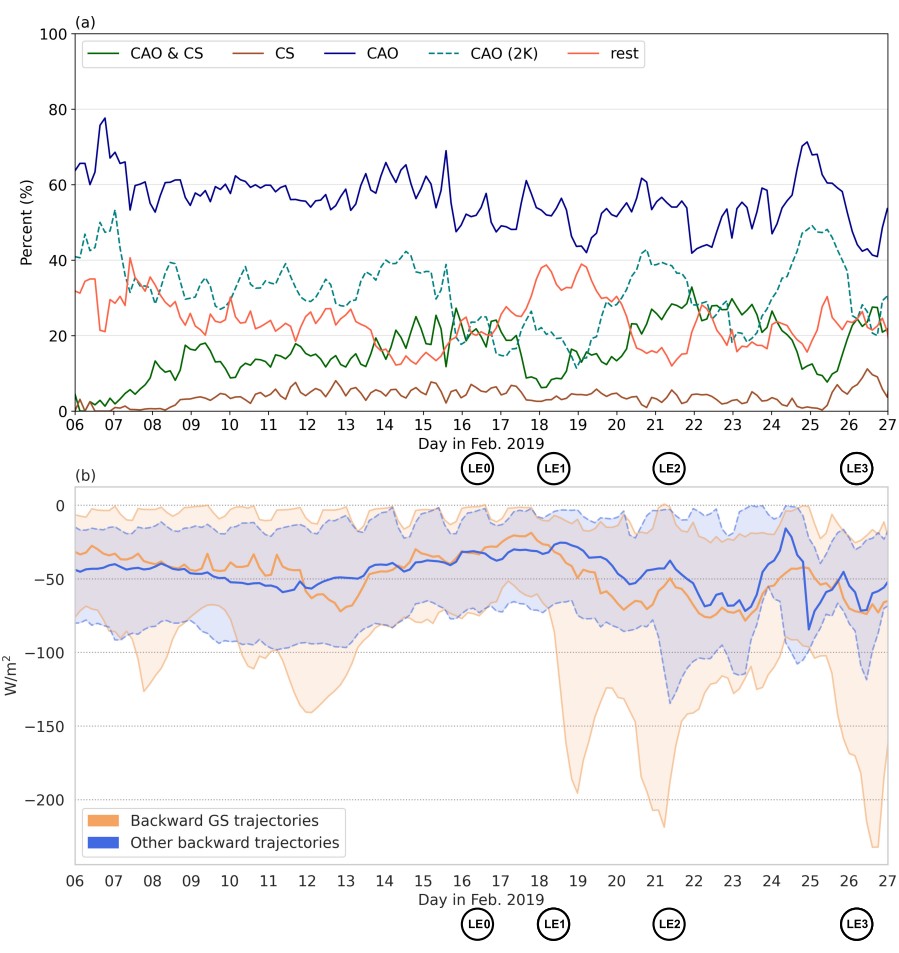

**Figure 7.** The properties of the air parcels during moisture uptakes. (a) The fraction of all moisture uptakes for NPVA GS trajectories at a given time, categorized by different regions: CAO - affected by the CAO event identified using the definition $\theta_{SST} - \theta_{850} > 0$ K, CAO (2K) - affected by the CAO event identified using the definition $\theta_{SST} - \theta_{850} > 2$ K, CS - cold sector of the cyclone, CAO&CS - identified as both CAO and the CS, rest - trajectories that are not CAO, CAO(2K), CS or CS&CAO, for all NPVA GS trajectories at given time step between 6 - 27 Feb. 2019. (b) Average surface latent heat flux of the - orange NPVA GS and blue - NPVA nonGS trajectories at a given time from 6 to 27 February 2019, the 90th and 10th percentiles are plotted as light orange and blue shading, respectively. LE0-3 refers to starting dates of cyclones from Fig.2

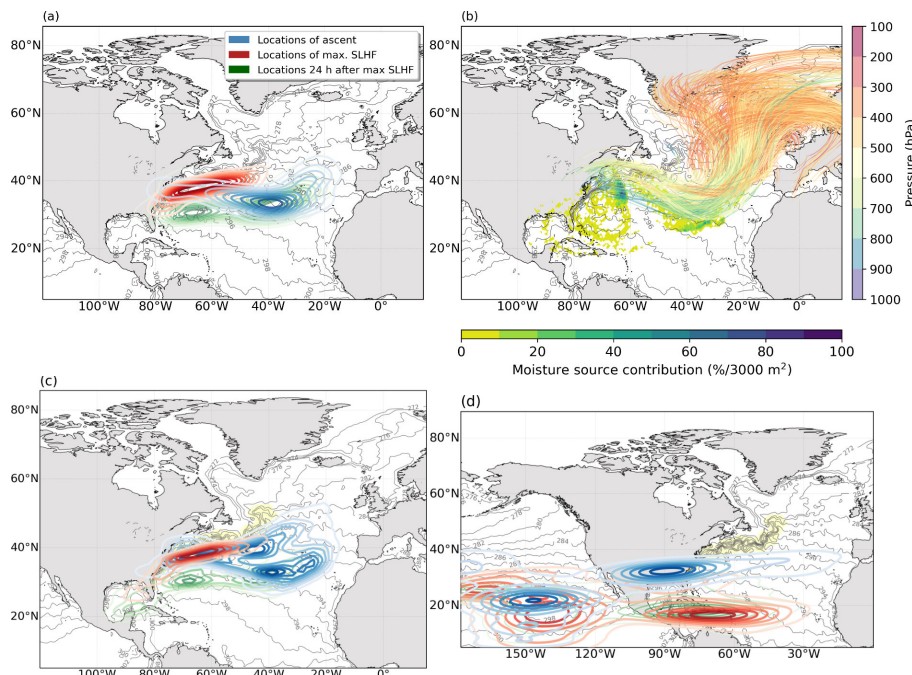

**Figure 8.** Kernel Density Estimation (KDE; using Scott's rule; Scott (2015)) of air parcels distribution at the time whem when maximum value of surface latent heat flux is found along the trajectory (red contours), 24 h hours later (green contours) and when they start ascending (blue contours) (a) for trajectories started on 24 February at 21:00 UTC, (c) for all the NPVA GS trajectories, (d) for all the NPVA nonGS trajectories from 20-28 February 2019. Contours represent 10% steps of the density of air parcels. (b) The pressure change of trajectories started on 24 February at 21:00 UTC during ascent, together with moisture sources contribution to total moisture content prior to ascent as in Fig.6 (e-f).



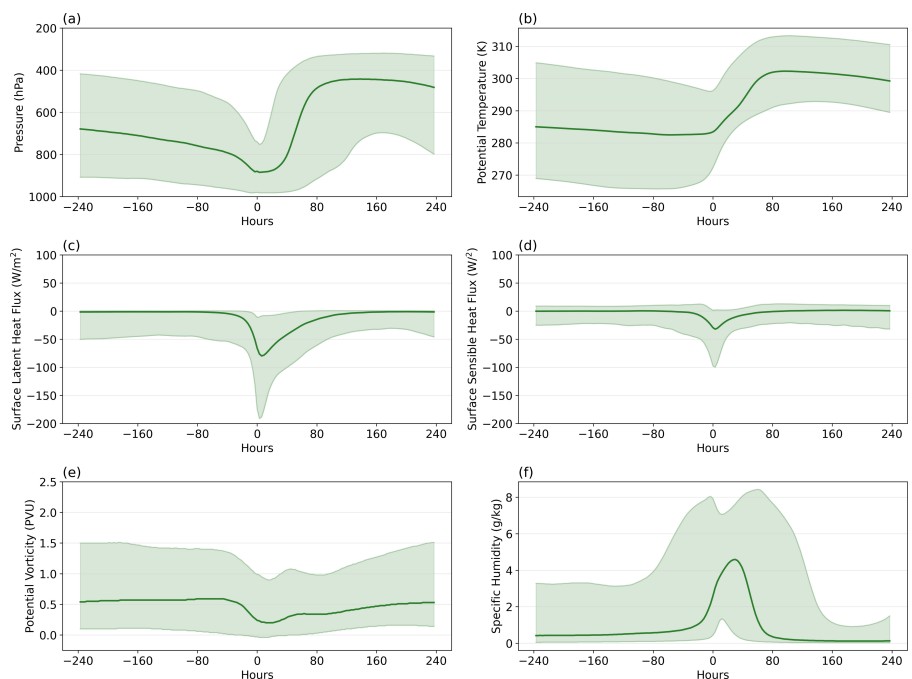

**Figure 9.** Temporal evolution of: (a) pressure, (b) potential temperature, (c) surface latent heat flux, (d) surface sensible heat flux, (e) potential vorticity, (f) specific humidity along GS NGPVA trajectories. Time 0 h refers to the start of the trajectory in the Gulf Stream region. The medians are represented as thick green lines and the 90th and 10th percentiles as light green shading.

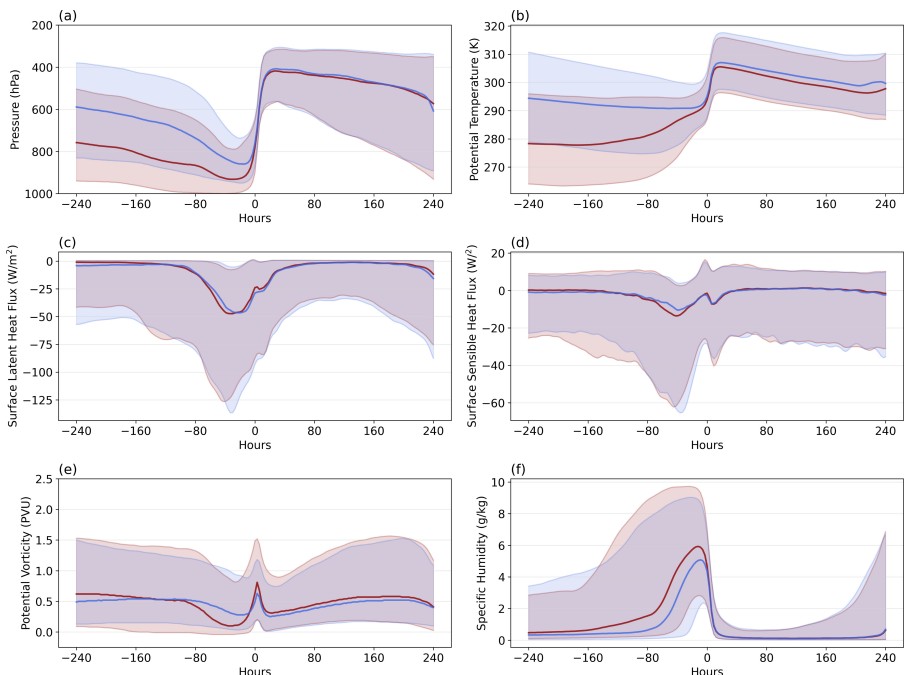

**Figure 10.** Temporal evolution of: (a) potential vorticity, (b) pressure, (c) surface latent heat flux, (d) surface sensible heat flux, (e) potential temperature, (f) specific humidity along GS NPVA negPV trajectories (red) and GS NPVA posPV trajectories (blue). Trajectories are centered on the time step with maximum latent heating (hour 0). The medians are represented as thick red and blue lines and the 90th and 10th percentiles as light red and blue shading.



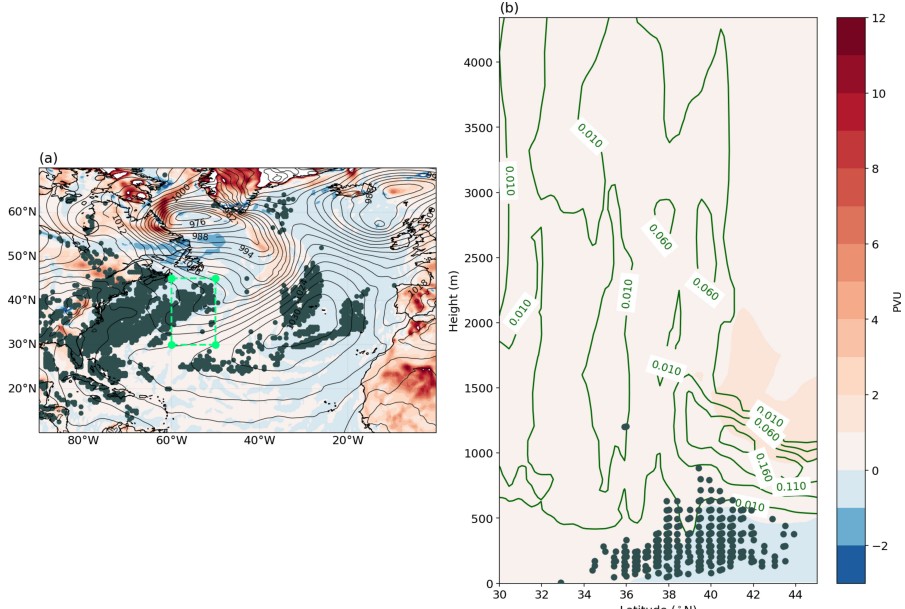

**Figure 11.** Vertical and horizontal distribution of negative potential vorticity over the area inside the green box in the plot (a) for 18 February 00:00 UTC. (a) Potential vorticity on the lowest model level and locations of air parcels with negative PV in the atmospheric boundary layer. (b) Vertical distribution of potential vorticity (shading) and liquid water content (green contours) averaged over the area between 55° - 60° W and 30° - 45° N, together with the location of all air parcels from the box that have negative PV in the atmospheric boundary layer on 18 February 00:00 UTC. The plots were created using ERA5 data featuring a 1-hour time resolution.

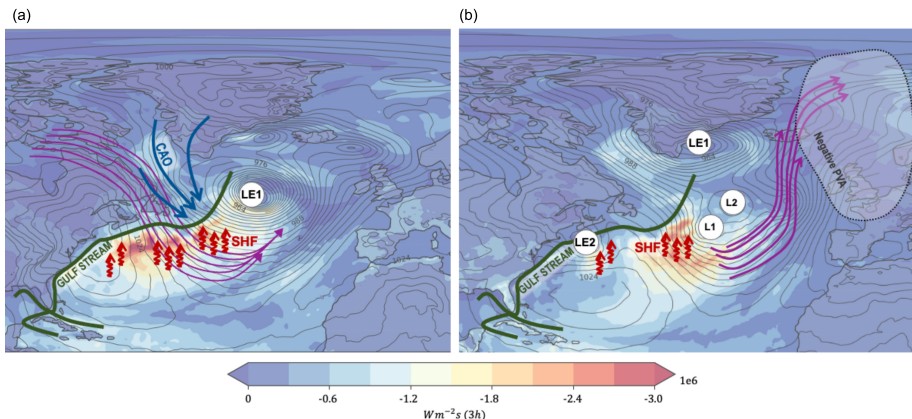

**Figure 12.** Schematic ilustrating the 'hand over' mechanism. Green line - Gulf Stream SST front, red lines - areas of intense sea-air heat and moisture exchange, purple lines - trajectories, blue lines - the advection of cold air. Consecutive cyclones are marked as LE1, LE2, L1, L2 (Fig.3). Gray contours represent surface pressure in in intervals of 4 hPa and shading the surface latent heat flux on 20 February 12 UTC (a) and 21 February 18 UTC (b) (accumulated over a 3h period).