# Peer review of "Linking Gulf Stream Air-Sea Interactions to the Exceptional Blocking Episode in February 2019: A Lagrangian Perspective"

_EGUsphere, 2023_

## Author Response (AR1)

**Reviewer 1**

First of all, we would like to thank the reviewer for their extensive comments and detailed analysis of our paper. We greatly appreciate it and we think that following those suggestions greatly improved our paper.

First, I would like to recognise that a lot of work has gone into this article. This is a large analysis and has required a lot of thought by the authors and there are many interesting lines of evidence presented. Furthermore, I think the methods the authors use are the correct ones for understanding these very important processes. Nevertheless, I feel the paper is confused and lacks a clear focus – do you want to show that CAO air is modified, incorporated into an extratropical cyclone, and then transferred to upper levels to help develop a blocking event? If so, then they need to show this happening much more clearly. The paper shows each of these steps individually to some extent, but there is no coherent trail of evidence that shows the whole process happening. Furthermore, it seems the authors are so focussed on the "handover" phenomenon that they do not acknowledge that a minority of air parcels (~23% maximum) are subject to this process. They also do not use the trajectory method to show this happening in "real time" around LE2.

Thank you for this overall perspective on our paper. Indeed we aimed to showcase the chain of processes including air mass transformations over the Gulfstream, i.e., air-sea interaction in a CAO, and the ascending airstreams, affecting upper-level blocking downstream. Consequently, we conclude that the moisture uptake feeding the WCB, highlighted by the hand-over mechanism and the sequence of cyclones, is key.

However, we agree that the way we presented the different perspectives in the initial version was confusing. Taking your and the other's reviewer comments into account, we substantially revised the manuscript in order to streamline the storyline. Most importantly, we now only focus our Lagrangian analysis on the backward trajectories originating from the block. Regarding your comment on our focus on a minority of the air parcels, indeed, we primarily only use the relatively small subset of trajectories originating from the block and interacting with the Gulfstream. This specific focus allows us to carve out dynamic evidence for our main objective: to understand how signals from air-sea interactions over the Gulf Stream are conveyed to the upper troposphere, which is why we chose trajectories that pass over the Gulf Stream. In the revised manuscript we now clearly state this focus and explain the subset used.

Our study builds on recent insights into the role of the Gulf Stream for downstream blocking formation. Specifically, existing research indicates that the absence of the Gulf Stream leads to reduced blocking events (Cheung et.al. 2023), decreased atmospheric river activity, and decreased intensity of ascending motions over the North Atlantic, influencing Rossby Wave dynamics (Ma et.al., 2023, preprint). In addition, several studies emphasize the intricate connection between the ocean and the atmosphere. They suggest that more detailed models with improved Gulf Stream representation show a more accurate frequency of blocking events (Athanasiadis et.al. 2022, Scaife et.al. 2011, Michel et.al. 2023). There's also evidence pointing to the Gulf Stream's role in shaping large-scale atmospheric dynamics (e.g., Joyce et.al. 2019). Thus, the importance of the Gulf Stream for blocking formation is well established. However, a mechanistic perspective on HOW air-sea interaction affects the upper-level flow is still incomplete.

We aim to uncover the mechanisms behind these observed phenomena from a synoptic perspective. Our goal is to determine how exactly the Gulf Stream's signal gets to the upper troposphere. While the case study presented here provides a potential mechanism, we do not claim it as the sole driver for block occurrences. We are more focused on understanding the potential influence of the Gulf Stream on a block's lifecycle, which might explain why there is a decline in block frequency and intensity when the Gulf Stream is absent in certain models. In the revised version of the manuscript we now make clearer that specific focus.

It would be great to see the trajectories plotted running through a CAO, being modified over the Gulf stream, ascending in the WCB of LE2 and finally ending up in the NPVA, but the authors do not do this.

In the revised paper, the mechanism is detailed in Figures 9 and 10. These figures showcase the uptake locations and the position of a cyclone at the time of uptake, typically south of Greenland. Notably, after the moisture uptake when trajectories ascend, another cyclone is present in the region which promotes the upward movement of trajectories. Thus, air parcels gain moisture when they arrive at low levels in CAOs induced by the first cyclone and a second cyclone triggers their ascent (as visible in figures 9b,c and 10b,c). We haven't explicitly demonstrated that these trajectories are WCBs. However, their average pressure, represented in color-coded pressure, highlights their ascent, along with the average location and timing of this ascent. We trust this provides a satisfactory response to the reviewer's query.

I am also worried that the way the analysis is presented overstates the role of CAO-GS "handover" mechanism in the overall makeup and development of the NPVA regions at upper levels. From my viewpoint, most of the trajectories (~77%) do not pass near the GS and are from a different origin (Gulf of Mexico for example) and therefore the influence of the implied low-level modification is relatively small. That does not mean that they are not important (I'm very willing to believe that they are) but the paper focuses so much on the ~23% of parcels that may have originated over the GS that they ignore much of the makeup of the NPVA. Again, I can easily believe that the ~20% of highly modified parcels that end up in the NPVA region can "tip the balance" between short-term ridging and persistent block development but the arguments are not presented in this way. The paper consistently focuses on the GS modification of CAOs as being the primary mechanism at work here whereas it is not the main mechanism.

As initially stated, we chose to concentrate primarily on these trajectories, even though they represent a minority. Our objective was to understand from a mechanistic point of view how air-sea interactions over the Gulf Stream can impact blocking and to outline the underlying pathway of this influence. We'd also like to acknowledge a minor error in the manuscript: the number 23 should have been 28. However, this doesn't significantly alter our focus, as our primary intention was to emphasize these specific trajectories. We revised the text in order to avoid the misunderstanding that these trajectories might be alone responsible for the blocking onset.

Another concern is some of the lack of checking of details around figure contents, labelling, and captioning. It is consistent with some of the lack of detail presented in the text itself, which may be due to there being too many hypotheses, arguments, and datasets in the

paper. Removing some of the unnecessary content would likely help clarify and improve this work significantly. Again, I would like to commend the authors for having done a tremendous amount of work here, but I would argue a third of it is not relevant. I recommend the authors review what they are trying to say in this paper and re-organise it to focus on the process they are interested in. The authors also need to give a much fairer/clearer reflection of the role of nonGS contributions in the overall makeup of the blocking event to give a better context to the role of the GS-induced "handover" mechanism.

We recognize that certain aspects of our original manuscript lacked clarity, and we appreciate the reviewer for pointing this out. In the revised version, we have tried to present these elements more coherently and enhance the labeling of the figures. Also, we removed the discussion of forward trajectories completely, in order to remove unnecessary content.

I recommend the authors carefully consider the comments below. Removing unnecessary analysis, adding some finer detailed analysis (e.g., showing trajectories starting in a CAO, being modified, ascending in LE2 and ending up in the NPVA) and providing a much stronger acknowledgement of the relative importance of the GS trajectories to the non-GS trajectories would make this a fascinating and highly important paper. As it stands, the paper does not really show the processes described. Instead, the paper infers a lot of the processes, and provides too much emphasis on the mechanism that accounts for only 23% of the phenomenon under investigation.

Specific Comments

I would suggest removing all the GS trajectory analysis and the DI, DH and CAO1 trajectory analyses for NPVA. You are interested in the trajectories that end up in the NPVA so just stick with those trajectories. Also, it is important to make sure you quantify those NPVA GS trajectories in context of the nonGS trajectories as that tells you what proportion of the GS trajectories influence the NPVA. This is something that the paper severely lacks i.e., the proportional contribution of NPVA GS trajectories, which is the minority of trajectories.

Thanks for this suggestion. We agree that focusing solely on NPVA trajectories greatly helps to streamline the manuscript and improve its clarity. In the revised manuscript, we limited our analysis to the 10-day backward trajectories originating from the NPVA objects. Additionally, we assessed the proportion of NPVA GS trajectories relative to NPVA nonGS trajectories.

Furthermore, I find there are also so many acronyms that it is hard to keep track of what they all are. I also think the CAO2 trajectories are the most important, followed by the WCB trajectories (and the nonGS trajectories, which are in the majority). Both CAO2 and WCB are strongly tied to diabatic processes anyway, which negates the requirement to show the DH trajectories. I would then remove the whole GS section at the end (Section 3.5). If you wanted to do the PV trajectories, then you should do it on the NPVA trajectories as I believe they are the most important ones. This will help focus the paper much more and remove a lot of unnecessary text.

We have incorporated the reviewer's suggestions into our analysis. We determined the fractions of each air stream for NPVA trajectories and conducted a detailed analysis. Additionally, the examination of negative PV in the trajectories has been relocated to the Appendix. Nonetheless, we chose to compute the proportion of DH trajectories. Recent

studies by Steinfeld et al. (2019 and 2021) highlight the significance of these trajectories for block strengthening and maintenance. It's important to note that a trajectory can qualify as DH without being a WCB or CAO. This is due to the fact that while the WCB provides a specific criterion for identifying ascending airstreams, not every ascending airstream meets this criterion. Furthermore, diabatic heating can take place during ascent related to extratropical cyclones even if CAOs are not present.

L60: Is it worth including the idealised study of Boutle et al. (2011) here too as they show the timescale for moisture adjustments in the boundary layer to be approximately 2.3 days for highly idealised situations and it fits in with the small timescale required for the "preconditioning" you mention.

We want to thank the reviewer for bringing this article to our attention and we decided to add it to the Introduction section of the paper.

L62-63: "However, the pathway of CAO influence on the upper-level flow is unclear" – not strictly true. These cold air outbreaks can be associated with a strong upper-level jet stream at their boundary with warmer sub-tropical air. This situation is seen frequently over the winter. The orientation of the jet stream in that situation (e.g., south-west to north-east) could potentially induce ridge building without any need for "preconditioning". I would simply suggest removing this sentence as it is misleading and the content of the paragraph before this point is good on its own. Conversely, if you want to keep it, then you need to relate it back to the rest of the paragraph beforehand e.g. "the pathway from CAO, through diabatic modification of that cold air, subsequent transport through the WCB of a cyclone and then impacting the upper level flow is unclear" (or something similar to that).

We followed the suggestion of the reviewer and removed the mentioned sentence.

L115-116: I'm confused by the last sentence. I think you probably need to add a figure showing how you have "visually" done this. Either as a response or as a supplementary figure. Also, the way it is written makes it seem like you're looking at the "distribution of cold sectors" not "the distribution of the SSHF and PV within the cold sectors", which is what I think you mean. Just needs a re-word and a small example as evidence.

Given that moisture uptake for trajectories is more closely linked to a CAO caused by a cyclone than to the cold sector, we chose to omit this part of the analysis in favor of simplification, aligning with the reviewer's recommendation.

L155-157: This is not clear as to what you've done to initiate these trajectories. "…equidistant grid of $\Delta x=100$ km and $\Delta y=25$ hPa vertically between 500 and 150 hPa within both NPVAs…" By Equidistant do you mean 100 km between each trajectory starting point or that there is a 100 x 100 km square area you start them within (I'm thinking it is the former)? Are the crosses in Fig 1. representative of the starting points of the back trajectories for the upper level NPVAs? If so, then you should state this in the text in section 2.2.1 and the Fig 1. caption to make it much clearer. If the above is true, then I think the black crosses in Figs 1e and 1g are the wrong way round (i.e., the shape of the NPVAs in Fig 1e match the distribution of crosses in 1g and vice versa). Please check this.

We initiate trajectories from an equidistant grid, with each grid box measuring 100x100km. This has been clarified in the revised paper at the respective location in the text and Figure captions.

Fig 1e: The minor NPVA is difficult to see. Could you change it to a clearer colour to contrast with the existing blue shading? Possibly a magenta or pink colour would show better.

The color of the minor NPVA has been changed in the revised manuscript.

L285: "On average, more than 23% of the NPVA trajectories interact with the ABL over the Gulf Stream region…" so, most air parcels have no interaction with the Gulf Stream region? Therefore, the main cause of the NPVA development could be argued to come from other sources. Doesn't that mean the mechanism you propose is not the most important one?

We acknowledge that our initial manuscript may not have clearly conveyed that we do not view this as the predominant mechanism for block formation. While most air parcels don't interact with the Gulf Stream, our intention was to demonstrate how air masses from the Gulf Stream might influence it. Through our case study, we aimed to illustrate a potential pathway of the Gulf Stream's effect on the block. Therefore, our primary objective wasn't to pinpoint the impact of NPVA GS trajectories, but rather to explore the processes taking place along these trajectories and how they are linked to the synoptic weather systems /cyclones over the North Atlantic.

L295-324: The central idea behind this paper is that CAO air is modified around the Gulf Stream, ascends in WCBs within major extratropical cyclones and then impacts the upper-level circulation that led to the intense blocking event. If that is true, then why are there a larger fraction of CAO trajectories that end up in the NPVA (NPVA GS CAO) than travel through the WCB (NPVA GC WCB)? Surely, the percentage of CAO parcels should be less than or (at most) equal to the number of WCB trajectories, otherwise how does all the excess modified CAO air get into the upper troposphere? If the percentages are calculated as a "percentage of the fraction of NPVA GS" trajectories, then this needs to be clearly stated as the current plot is misleading by making it seem like CAO modification in more important than it actually is.

Our dataset incorporates a specific ascent criterion. The WCB, which represents an ascent of 600 hPa within 48 hours, sets a clear, well-established benchmark for ascending airstreams within a cyclone (e.g., Madonna et al. 2014). Many trajectories show an ascent of 500 hPa (e.g., moving from around 900 hPa to 400 hPa), but they might not meet the WCB criterium even though they are still part of the extratropical cyclone. In our case study, we apply an ascent criterion of a 500 hPa difference over a 10-day backward period. This ensures all trajectories ascend, though at different rates. Figure 4 illustrates this difference: NPVA GS trajectories mostly ascend within a 3-day backward period, while nonGS trajectories have a longer and more gradual ascent.

It's worth noting that while many trajectories qualify as CAOs, only some ascend quickly enough to be labeled as WCBs; others ascend a bit more slowly. Additionally, we detect CAOs only in the boundary layer (with a pressure greater than 800 hPa). Given that trajectories begin from pressures between 500-150 hPa, their ascent can cover a 400 hPa difference.

L321-323: "Moreover, there appears to be a connection between the increases in the fraction of WCB trajectories and the growing number of trajectories defined as GS CAO NPVA (blue and teal in Fig.3b; Tab.1)." This is very vague and I'm not sure I agree with the statement. Where exactly are they "connected"? Suggest removing or expanding.

In the revised version of the manuscript, this sentence is removed.

L325-331: I disagree with this; the DI trajectories are in the minority for a lot of this. Also, in general, they only account for <20% of trajectories in the NPVA i.e., lots of air parcels from other sources. Furthermore, their presence in the NPVA may be a result of them descending from the upper troposphere to the mid/lower-troposphere and then ascending back above 500 hPa without ever interacting with the surface. Moreover, the ABL will be very shallow where the DIs are active (quite likely below 800 hPa). I think you're inferring a lot here without showing any evidence. I don't think this paragraph should be included unless you can show DI parcels that have specifically interacted with the ABL and ended up in the NPVA.

We've made changes to this section of the manuscript but chose to still refer to the DI trajectories. Although they comprise only 2% of all NPVA base trajectories, given the distinct nature of DIs and their potential impact on the atmospheric boundary layer, we believe they offer valuable insights. Including them could benefit other research and shed light on the intricate processes occurring over the Gulf Stream.

Recent research underscores the significance of DI's. Raveh-Rubin (2017) showed that the intense evaporation caused by DIs introduces newly evaporated moisture into the PBL. When this moisture is lifted and transported, it can contribute to downstream precipitation, as pointed out by Winschall et al. (2012). Moreover, the transport of air masses from the PBL to the free troposphere is predominantly carried out by cyclones and fronts via the upward movement in WCBs, as documented by Cotton et al. (1995). Therefore, DIs can possibly influence cyclogenesis and frontogenesis. Adding to this, Catto and Raveh-Rubin (2019) found that all frontal types, when coexisting with a DI, display stronger temperature gradients, cover a more expansive area, and generally have enhanced precipitation levels.

Figure 5a - the caption describes "Contributions of accumulated moisture uptakes to moisture present…" first – are these words referring to the "fractions explained" axis? If so, then that axis should be defined on the left with the "number of uptakes/day" the right (as you discuss fractions explained first). Also, the "fractions explained" label should be "accumulated moisture fraction of total moisture present (%)" to reflect the caption. Furthermore, I think you need to define what a "moisture uptake" is as the term is arbitrary. If it were quantified as an actual mass of water, then we could see how important the process is. If this moisture uptake were accumulated across all NPVA GS and NPVA nonGS trajectories, then we could see which has a bigger impact on the total NPVA moisture.

Figure 5b – I think this figure is misleading as it does not indicate the absolute values. If I look at the bars, both the NPVA GS and NPVA non-GS fractions add up to 100% i.e., they are relative to themselves. If NPVA nonGS accounts for 77% of all NPVA trajectories and NPVA GS accounts for 23% then the impact of the NPVA GS trajectories is significantly diminished. The plot currently implies that the NPVA nonGS and NPVA GS are equally important, which

cannot be true if there is a 77/23 split. These fractions need to be presented as a percentage of all NPVA trajectories to gauge relative importance.

We concluded that Figure 5 might have presented a potentially misleading interpretation of our findings. Given the introduction of new figures in the paper, we chose to exclude it from the revised version. We trust that the additional figures more effectively communicate our main emphasis: the distinct regions where NPVA GS trajectories and NPVA nonGS trajectories primarily acquire moisture, and the observation that NPVA GS trajectories predominantly pick up moisture in the CAOs, specifically those occurring behind cyclones and over the Gulf Stream.

The reviewer's feedback on the labeling in Figure 5a is valid, and we acknowledge that the depiction could have been clearer. The suggested captions indeed offer a more precise representation of the data visualized in the plot. To clarify, moisture uptake refers to the increase in specific humidity between time steps.

Regarding Figure 5b, the reviewer's interpretation is right. However, although the plot seemed to assign equal significance to both datasets, our intention wasn't to imply that both are equally crucial for block development. Rather, we aimed to highlight the distinct characteristics of trajectories interacting with the Gulf Stream.

L370 and Fig 6c: Suggest changing "are related" to "are likely related" as you don't show what level in the atmosphere these uptakes occur. If you could plot the pressure height at which the maximum uptake of each trajectory occurs (and where that is) then you could say "are related". It is not inconceivable that a layer of moist air at e.g., 850 hPa was advected into the domain you're looking at from a lower latitude before the last point in the back trajectory. The moisture would therefore "look" like it came from the Gulf Stream but could have originated elsewhere in the sub-tropics many days before. The inclusion of the word "likely" negates this by acknowledging that you're not 100% certain.

This section of the manuscript has been updated in the revised version. Additionally, in Figures 9 and 10, we've illustrated the locations of the most pronounced moisture uptakes (with figures for all case study time steps available in the Supplementary material). We trust this provides clearer insight.

L368-378 more generally: Just because the CAO index is co-incident with the moisture uptake field does not mean that the two are related. Your argument would be much better if you plotted the NPVA CAO2 GS trajectories' moisture uptake frequency and contributions so that it could be compared to both the NPVA GS and NPVA nonGS to see the relative importance of the CAO modification process.

We chose not to center our analysis exclusively on NPVA GS CAO trajectories, as the main emphasis of the paper is on NPVA GS trajectories, and not all of them are specifically linked with CAOs. However, in response to the reviewer's suggestion, we computed the total moisture accumulated for each time step of the case study, distinguishing between NPVA GS trajectories identified as CAO and those that weren't (see Fig. 7c). From this, it's evident that NPVA GS trajectories identified as CAO absorb more moisture than their non-CAO counterparts.

Fig 7/L379-394: The CS analysis doesn't add much to this, and I think this figure would be much improved by removing all lines but the CAO2 line. That way you could clearly see the impact of LE1 and LE2 for creating the conditions for large moisture uptakes. I also don't think lines 386-394 are necessary and can be removed.

We followed the suggestion of the reviewer.

L393-394: "It is worth noting that the episodes of extreme SLHF are significantly larger in magnitude for the NPVA GS trajectories than for the NPVA nonGS trajectories (Figure 7b)." – I think it is important to note here that large heat fluxes are the result of a large difference in characteristics between two different media. The heat fluxes are high because there is cold, dry air lying over a relatively warm, moist ocean. If the fluxes remain high, then the air must still be cold and dry. Low heat fluxes imply that the imbalance is much smaller (i.e., the air is relatively warmer and moister, which reflects the underlying surface conditions). So, while the SLFH may be more extreme in the NPVA GS (and possibly NPVA COA2 GS) trajectories, the overall moisture (and heat) content of the NPVA nonGS trajectories might be higher (and more important) in absolute terms – especially if there are a lot more NPVA nonGS trajectories than NPVA GS trajectories. I think you must employ some sort of weighting here to account for both the overall number of trajectories in each group and their total moisture content. A more modified air parcel may contain less moisture than an unmodified air parcel.

We acknowledge the reviewer's point that we might have placed excessive emphasis on the significance of high surface latent heat fluxes. Nonetheless, elevated surface heat fluxes can suggest that cyclones leading to these high fluxes are typically deeper than average and more frequently undergo rapid intensification, as found by Tilinina et al. (2018). Such fluxes also influence the lower troposphere, altering low-level baroclinicity and, consequently, affect the dynamics of the cyclones that induced these fluxes initially. Additionally, surface fluxes in the western boundary currents can stimulate local tropospheric moisture recycling by triggering convective processes. Collectively, these factors indicate an unstable environment, likely linked with CAO, characterized by significant moistening and heating events.

L395-396: "Our results are in agreement with other studies (e.g. Papritz and Grams, 2018; Aemisegger and Papritz, 2018; Hawcroft et al., 2012), indicating that CAOs play an important role in the water cycle of cyclones" I think you could actually show how important the role is by quantifying the contribution of CAO2 moisture to the total moisture in the NPVA if you follow through with the suggestions I've made above. That would be a neat and significant result (even if the value seemed relatively small).

We believe that the newly added Figure 7 (alongside Figures 9 and 10 and the supplementary materia)l, underscores the idea that the most prominent moisture uptakes predominantly occur in CAOs, as especially evident in Figure 7c. A clear trend in Figure 7b suggests that uptakes within the CAO regions are more prevalent than those outside, especially for NPVA GS trajectories. Moreover, a comparison of the average moisture uptake intensity (specific humidity increase over 3 hours) inside and outside CAO regions for NPVA GS trajectories reveals that uptakes within CAO regions tend to be more intensive - particularly when we have intensive cyclones active in the North Atlantic.

L413: "and intensifies in the region of strong CAO left behind by LE1 (Fig.3b). The already moistened air is then fed into the ascending airstream of the LE2 cyclone" – I disagree with this. The plot shows the cyclone intensifying in the air behind the CAO region. Assuming a westward tilt of the system, it should be developing because of conditions upstream at upper levels. This statement appears to be suggestive of the CAO region causing the intensification, which is not actually shown. You also do not show that the "already moistened air" (assuming you mean modified CAO air) has been fed into the ascending air stream of LE2. You would need to show the specific trajectories that follow this path for me to be convinced this statement is true.

We trust that Figures 9 and 10, along with the Supplementary material, provide a clearer illustration of our main points. Additionally, it seems our explanation may not have been precise enough. What we observe is trajectories passing through a CAO created by the passage of one cyclone, and subsequently being elevated into the upper troposphere by another cyclone. We aren't suggesting that the cyclone intensifies due to the CAO. Instead, we observe a spatial overlap of the CAO left behind by Cyclone L1 and Cyclone L2. This suggests that the robust air-sea interactions during the CAO might influence the intensification of L2, though quantifying this specific point is beyond the scope of our analysis.

L416-429: Again, I think there is mainly inference here rather than proof. First, the statement on the NPVA GS "Trajectories begin their ascent into the upper troposphere on average 3.5 days after reaching maximum SLHF…" when combined with the statement in the previous paragraph stating "The ascent occurs approximately 54 hours after the maximum SLHF values" is indicative of the NPVA nonGS trajectories ascending well ahead of the NPVA GS trajectories, which is unsurprising because the high SLHF values imply that the air is still relatively cold and dry and needs more modification before it can be lifted. Also, when are Figs 8a and 8c representative of? There's no information on when the maximum SLHF was occurring relative to the development of LE2. From Fig 3b, it is clear LE2 was very well developed by 2000Z on 21-02-2019, so working back from 2100Z on 24-02-2019, 2.6 days to ascend and 3.5 days to accumulate moisture means that these parcels were (on average) in a CAO approximately 6 days before and suggests they were associated with the modified CAO before the passage of LE1 (as can be seen in Fig 3a). So, it is unlikely that your proposed "hand over" process is working on the timescales you suggest. Again, if you were to plot the trajectories specifically associated with LE2 and they were clearly from the modified CAO following LE1 then I would believe your argument. As it stands, your own results suggest your proposed mechanism does not work on the timescales you propose.

Figures 9 and 10, complemented by the Supplementary material, aim to offer a clearer representation of our arguments. On reflection, we acknowledge that our initial explanation might have lacked clarity. We've observed trajectories navigating through a CAO, which forms after one cyclone passes, are later lifted to the upper troposphere by a subsequent, separate cyclone. We are not suggesting that the presence of the CAO directly leads to the intensification of the cyclone. Rather, our observations indicate a concurrent positioning of the CAO, left in the wake of Cyclone L1, with Cyclone L2. This pattern hints at the possibility that pronounced air-sea interactions during the CAO phase could potentially impact the intensification dynamics of L2. However, a deeper dive into this particular inference isn't the central thrust of our analysis.

L436-437: "NPVA nonGS trajectories tend to have a higher number of uptakes compared to NPVA GS trajectories, accounting for 75% of all NPVA trajectories." I think this is a really important point, i.e., that most of the moisture uptake is not from the Gulf Stream region. Therefore, most of the air parcels reaching the NPVA region are not arriving there because of the "hand over" process.

Our main focus is on NPVA GS trajectories. We are not suggesting that these trajectories make up the majority of air parcels reaching the NPVA or that the 'hand over' mechanism is the only mechanism influencing the block formation. Rather, our goal is to show that this mechanism, or a sequence of cyclones, creates a pathway for air masses interacting with the Gulf Stream to later rise into the upper troposphere.

L474: "Gulf Stream can be attributed to the influence of CAOs" – how have you shown this for these trajectories? Can you show that these trajectories/parcels were part of a CAO and then modified? As it stands you have only shown that these GS NPVA trajectories change their properties with time and have not shown that they were part of a CAO. You need to show the evidence before you can make this statement.

Due to the exclusion of GS trajectories from the article, per the reviewer's suggestion, this section has also been omitted.

L507-508: "as evident from the temporal changes of sensible heat flux in the two types of trajectories. Indicating that processes occurring during CAOs may be responsible for the decrease of PV in the atmospheric boundary layer" – again, this is inferred but not actually shown. You need to show that these specific trajectories came from a CAO for this statement to be valid.

We determined that 85% of negative PV values occur in the CAO region. This information has been included in the Appendix where we've now shifted the analysis of negative PV in the boundary layer.

L509-521: I'm not sure what the purpose of this paragraph is. You discuss a "cold sector" but there isn't really a cyclone nearby. I could believe a trailing cold front lies in the vicinity of the box in Fig 11a, but it is very remote from the parent cyclone. All the sentences on stratiform clouds are being used to infer the presence of a cold sector – why not just plot the vertical temperature field and show there is a thermal contrast? I think the paragraph is trying to explain the reason behind the negative PV, but there is a lot of inference here without showing the actual processes (e.g., evaporative cooling) happening. Finally, there is an inference that the cloud is "stratiform" in nature, but that is not shown conclusively either.

We initially introduced this analysis within the main body of the article, but given its supplementary nature, we opted to shift it to the appendix. This analysis provides insight into the ongoing discussions about the role of negative PV in the lower troposphere and the debate on the conservation of PV along the WCB, as highlighted by Methven (2015). We've chosen not to delve extensively into this particular topic in the main content. However, for added clarity, we have included a plot showcasing the averaged vertical cross-section of potential temperature, emphasizing the unstable environment of these air parcels. We believe that for an appendix, such an analysis is sufficiently detailed.

L530-531: "positioned ahead of the cold front", what cold front? It is not shown on the figures. It is important to include where this feature is in the plots (or where you think it is at the very least).

We've added a line to Fig.A2 indicating the location of the cold front (represented by a bold dashed line) and the subsequent "trailing" cold front (denoted by a thinner dashed line).

L532-535: "Considering the handover mechanism's predominance in our case study and the findings presented in Figure 10, it is reasonable to expect that their PV will increase within the next few hours, and they will be carried upwards into the upper troposphere by the ascending airstream of cyclone LE1 (Fig. 2)." – Again, you make this statement and then do not show the process that is central to it. Why not show the trajectories for this whole set of steps (cold air, modification, lifting in cyclone LE1 – or should this be LE2?) to make it completely clear that what you say is true. I haven't seen any conclusive evidence to support this statement.

We understand that some things might have been overstated in the initial version of the manuscript and we hope that in the revised one Figures 9 and 10 clearly depict the process we're highlighting. Considering that 85% of negative PVs are observed in the CAO and that these trajectories ultimately reach the upper-level NPVA, it's evident they undergo ascent. From the data indicating negative PV in the lower troposphere, primarily within the CAO, and insights from Figures 9 and 10, we deduce that these trajectories likely ascend under the influence of the next cyclone. We believe our analysis sufficiently presents this point for an Appendix and have chosen not to delve deeper.

L547-562: I disagree with this paragraph given the timings I have noted above i.e., the timescales for your processes are indicative of air originating from the cold air outbreak that preceded LE1 as it takes ~6 days to modify the air enough to rise in the LE2 cyclone. The lack of a trajectory analysis showing these specific processes means that the mechanisms you describe have not been fully proven to occur. I accept they may be occurring (and it would be fascinating if they are) but you really have not provided enough evidence to support that.

We trust that the updated manuscript, complemented by new figures and expanded analysis, offers a comprehensive understanding and evidence of these processes.
We show now that the moisture uptakes and ascents happen when different cyclones are present in the North Atlantic, primarily south of Greenland. We also show that the uptakes tend to occur in the CAO regions, developing in the wake of one cyclone, while the ascent occurs ahead of another cyclone. Those results are mostly shown in Figures 9 and 10 (and supplementary material), but also 3, or by looking and probability density distribution of time difference between the strongest uptake and the start of ascent.

L573-577: This whole paragraph is central to the point I am making. Why do the parcels originating outside the North Atlantic matter less than those that originate over the North Atlantic? Given, by your estimate, they constitute 77% of the make up of the NPVA region then they are surely the most important. The last sentence, "remote sources of moisture do not appear to be relevant for the air masses ascending into the block within the extratropical cyclones that formed in the North Atlantic in February 2019." That is a true statement, but these air masses are VERY relevant for the block overall as they constitute the majority of the

NPVA region from your trajectory analysis. The sentence seems to discount non-extratropical cyclone processed air as irrelevant, which it clearly is not.

We've made revisions to the manuscript to more explicitly outline our objectives. We trust that this updated version clarifies why we put so much emphasis on a subset of all NPVA base trajectories.
As emphasized earlier, we don't assume that NPVA GS trajectories are the primary drivers of block development. Our focus on them is driven by a desire to trace the route that air masses, which interact with the Gulf Stream, take to reach the upper-level block. Drawing from insights in studies such as Cheung et al. 2023, Athanasiadis et al. 2022, and Scaife et al. 2011, we hypothesize that this route might be a channel through which signals from SST changes are transported to the upper troposphere. In this context, extratropical processes become especially relevant for air masses interacting with the Gulf Stream, but less so for those that don't.

Once the above has been considered, I recommend you return to the last paragraph of the introduction and clearly state your aims as a list of bullet points while being careful to make sure they tie in with the final outcomes of the paper. Stating the aims much more clearly would help this paper considerably.

In the revised version of the manuscript, we've opted against using bullet points. Instead, we've refined the text to articulate our objectives more clearly. We trust that this adjustment provides better clarity regarding the intentions of our study.

We have addressed the subsequent technical corrections, except in instances where we've omitted certain sections of the manuscript, primarily those related to GS trajectories, based on the reviewers' recommendations.

Technical Corrections

L18: Change "ascent" to "ascend"

L109: I think "minima" should be "minimum" in the context you've used.

L119: Change "Block" to "block". I can understand why you use upper case for "European Block" as a named thing, but in this case, it isn't necessary unless you say "European Block" instead.

L150: You need to define what "GS" is here (I'm assuming Gulf Stream, but this is the first instance of the acronym being used and should be stated).

L199: Change "in, line" to ", in line".

L232: Change "Europe has" to "Europe had".

L236: Change "warm temperatures" to either "high temperatures" or "warm conditions". Temperature can be high or low not warm or cold (analogous to that, you wouldn't get wet and dry rainfall).

L262: Change "reinforces" to "reinforced".

L266-267: "…from the west…" might read better than "…from western direction…".

L268: Should this be Fig. 3c not 1c?

Fig 4 caption: You need to define what (a) and (b) are in the caption. Also, LE1 is labelled around 20th February in (a) whereas it is labelled between 18th and 19th February in (b). This should be corrected and consistent in both figures.

L283-295: Is it worth mentioning here that there are fewer trajectories initiated at lower levels than upper levels? To first order, you might ask "why do the percentage of NPVA GS trajectories not match the GS NPVA percentage?" given they should be starting / finishing in the same place if the tracers are to be believed (and therefore the same number). Just some clarification on why NPVA GS does not equal GS NPVA would be useful (1-2 sentences maximum). Again, this is coming from a non-expert for trajectories so the clarification may be useful to others.

L286: Should it be Fig 4b not 3b?

L302: Are those percent values for the CAO>0 or CAO>2K setups? You should clarify which. [In fact, what you say on L304-305 regarding this should be stated at L302]

L304: Do you mean Fig 4 instead of Fig 3 at the start of the line?

L307: after "atmosphere" include "(i.e., diabatic heating)" just to link better to the next sentences where you stop saying CAO and instead describe diabatic heating.

L309: Fig 4 not Fig 3?

L310: You can just say "exceeds 30%" as there's no need to give a range when quoting and exceedance.

L311: Suggest changing "contrast to a larger fraction obtained by Yamamoto et al. (2021) ~51,8%" to "contrast to the 51.8% obtained by Yamamoto et al. (2021) …"

L315: I don't think "therefore after the application of the ascent criterion" is required. Please delete.

L320: Should "revolution" be "resolution"?

L322 and L324: Again – should these be Fig 4 not Fig 3?

L325: Change "undergoes" to "undergo".

L339: Again, Fig 4 not Fig 3?

L409: Should this be Fig 8a not 6a?

L410: Should this be Fig. 8a not 6b?

L435: Should be Tab.2 not Tab/2.

L466: Change "have passed" to "has passed"

L503: "(Fig. ??a, b)" – please correct this.

References

Boutle, I.A., Belcher, S.E. and Plant, R.S. (2011), Moisture transport in midlatitude cyclones. Q.J.R. Meteorol. Soc., 137: 360-373. https://doi.org/10.1002/qj.783

**Reviewer 2**

We deeply appreciate the comprehensive feedback and analysis suggestions provided by the reviewer. Your insights have been very valuable, and we believe that integrating your suggestions has significantly enhanced the quality of our paper.

Firstly, we'd like to note that in response to the feedback from reviewer 1, we've made substantial modifications to the manuscript. This primarily involved omitting the analysis of GS trajectories and narrowing our focus to NPVA trajectories. We believe these adjustments have streamlined our arguments. Consequently, certain comments might not appear directly addressed due to the removal of those sections. Nevertheless, we trust that the reviewer will view these changes as positive improvements.

Review of 'Linking Gulf Stream air-sea interactions to the exceptional blocking episode in February 2019: A Lagrangian perspective' by Marta Wenta, Christian M. Grams, Lukas Papritz and Marc Federer

General comments

The paper constitutes a very complete piece of work tackling one important outstanding problem in dynamical meteorology, namely the connection between atmospheric blocking and surface processes and, in particular, the connection between the Gulf Stream, cold-air outbreaks, extratropical cyclone and blocking in the North Atlantic region, determinant of the European weather. The work relies on a variety of methods of analysis including cyclone tracking, the identification of negative PV anomalies and cold-air outbreaks, and Lagrangian trajectories. This work is fully in the scope of Weather and Climate Dynamics.

I give below a list of specific and technical comments that I believe can improve the paper by making it more understandable. Once these comments are considered I will be able to fully recommend the article for publication in this journal.

Specific comments

L25-26: 'cold surges' I can see that some of the papers referred to here do talk about cold surges or long-lived cold surges, but are these the same as 'cold spells'. In my opinion 'cold spell' is more appropriate as it does not imply the motion of cold air, but the effect of the persistence of large-scale conditions over a given region for a sufficiently long period of time.

We have modified it accordingly.

L56-57: 'the warm sector of the consecutive cyclone' Is there a more specific definition of 'the consecutive cyclone'? Is this any consecutive cyclone or are the authors thinking here of a specific flow configuration? From reading the paper I would say that it's better to talk about 'consecutive cyclones' in plural and that it's not always obvious which cyclone could be named the consecutive one.

In the updated manuscript, this point has been elaborated upon for greater clarity.

L67-71: I was not quite sure what was the argument around the wintertime poleward displacement of the jet stream and eddy heat fluxes. Is it the intensity or the position of the eddy heat flux that has the most influence on the position of the jet stream?

Both the intensity and position of eddy heat fluxes play a role in influencing the position of the jet stream. As demonstrated by Novak et al. (2015), a northern jet location in the North Atlantic is typically linked with periods of heightened meridional eddy heat flux over the Gulf Stream region. Additionally, O'Reilly et al. (2016) determined that intensified eddy heat fluxes along the Gulf Stream often precede large-scale upper tropospheric wave-breaking and the initiation of European blocking. Therefore, it's conceivable that an increase in frequent, intense eddy heat flux events over the Gulf Stream region would correlate with both an increase in European blocking frequency and a poleward shift of the eddy-driven jet.

L147-148: In the list of traced variables along trajectories there are five that are single-level. However the trajectories are located in three dimensions. How is the assignment of the single-level variables to trajectories made? Is there a criterion on the vertical distance from the parcel to the surface (single-level) or is the assignment made based on the horizontal position of the parcel only?

The assignment of surface variables is based on the horizontal position of an air parcel, traced directly beneath the trajectory. Therefore, when evaluating fluxes beneath trajectories and similar metrics, we only take them into account if the trajectory is near the surface, specifically at pressures greater than 800 hPa.

L194-196: The CAO index is based on 850-hPa potential temperature. Is there any condition on the level of the parcel as the potential temperature vertical gradient will be very different for two parcels with the same potential temperature difference but at very different pressure (altitude).

Indeed, the vertical gradient can vary for two air parcels with identical sea-air potential temperature differences. However, the primary goal of the CAO index is to identify air that is colder than the sea surface temperature, indicating it's convectively unstable in relation to the sea surface. The CAO index accomplishes this by contrasting the temperature of an air parcel in thermal equilibrium with the sea surface to one in the CAO boundary layer. This comparison is done only for the air parcels having pressure higher than 850 hPa. It's also worth noting that CAO boundary layers are typically neutrally stratified. In other words, the potential temperature is consistent throughout the boundary layer's depth, making vertical stability not particularly relevant, as observed in studies like those by Brümmer (1996) and other CAO observational research.

L263-264: 'The advection of cold air behind the cold front of LE2 resulted in another strong surface evaporation event…' How is the causality being assessed? Or is it that the CAO index is somehow being used as an indicator of strong surface evaporation?

The CAO index highlights a significant temperature contrast between the ocean surface and the atmosphere. It's well-documented that areas with CAO conditions often experience extremely high upward surface latent heat fluxes. For instance, studies by Tilinina et al. (2018) and Papritz et al. (2015) have emphasized this association. During wintertime CAO events, turbulent fluxes can even exceed 1000 W/m2, as illustrated by Bigorre et al. (2013)

and Marshall et al. (2009). Thus, consistent with the findings from Papritz et al. (2015) and Papritz and Grams (2017), the CAO index effectively signals areas of intense surface evaporation.

L323: I'm not convinced about the connection between increases in the fraction of WCB trajectories and increases in the number of GS CAO NPVA trajectories. I can see that there is coincidence to a certain degree but I would not say that this is systematic. Or perhaps I'm looking at this in the wrong way. Further explanation would be welcome. A similar comment is valid for L338-339.

This analysis has been modified in the manuscript. In the revised version of the article, we've omitted that figure because it wasn't conveying essential information. What we aim to highlight, and what is now more apparent in Figure 7, is the correlation between periods of intensive cyclone activity in the North Atlantic (specifically from 19-23 February) and an increased number of moisture uptakes occurring within CAO regions. Furthermore, these uptakes within CAO regions tend to be more intense than those outside of them. This underscores a potential linkage between CAOs and cyclones.

L383: In what cases are the cold-sector uptakes not associated with a CAO? Does this tend to occur at higher latitudes where SST is climatologically coder?

The analysis has been updated. Generally, moisture uptakes can occur, and often do, within the CAO resulting from the cold air advection associated with the passage of a cyclone. According to various definitions, this might not always meet the cold sector criteria, yet there remains a distinct air-surface temperature contrast as the cyclone passage resulted in an advection of cold air over the warmer water surface.

L387: I'm not sure I'm looking at this in the right way. From Fig. 7 the proportion of CAOs peak between 14 and 15 Feb and the SHLF is minimum on 13 Feb but LE0 intensifies rapidly only until 16 Feb. Perhaps it would help to mark these events directly in the figure using vertical lines.

In response to the Reviewer's feedback, we've updated the figure and our statements for better clarity. The updated figure shows the relationship between events more clearly. Specifically, NPVA GS trajectories experience much stronger fluxes. High surface heat fluxes, as highlighted by Tilinina et al. (2018), suggest that the cyclones causing these intense fluxes are typically deeper and intensify faster. These fluxes also change the conditions in the lower troposphere, moistening and heating the atmospheric boundary layer. Furthermore, these conditions point to an unstable environment, often found with CAO events.

Regarding Figure 7, we see the timing issue you mentioned. The high number of CAOs between 14 and 15 Feb and the lowest SHLF on 13 Feb come just before LE0's rapid intensification on 16 Feb. To make this clearer, we've added vertical lines to the figure to show these key moments. This should help readers see and understand the timing of these events better.

L391-293: If I'm following the argument, the idea here is that the CS of the cyclone should be completely covered by a CAO region. Since most uptakes take place in CAO regions in comparison with the cyclone's CS regions, it follows that the uptakes must take place in

CAOs that are not associated with that particular cyclone and therefore they must be associated with a preceding cyclone. If this interpretation is correct, then you might want to add 'preceding' in L392 "...occur in the CAO induced by the 'preceding' cyclone rather than…". Even if my interpretation of your argument is correct, it might be worth elaborating in your explanation. How are cold sectors assigned to a given cyclone? Is this done using a neighbourhood of a given size (e.g. a circular region of a given radius) around the cyclone centre? How big is this area compared to the CAO regions and does the area identified as the cold sector depend on the definition of the cold sector? Are the cold sectors truly a subset of CAO regions?

In the revised manuscript, we've simplified the analysis concerning NPVA GS trajectories. We observed that these trajectories often pass through a CAO, caused by one cyclone, which might occur behind it and not necessarily in its cold sector. Subsequently, they need another cyclone to rise into the upper troposphere. Thus, their moisture source might often be from a CAO behind the first cyclone, rather than its cold sector. To conclude, in the revised manuscript we decided to omit the analysis of air parcels presence in the cold sector.

L403: This comment is related to the comment to L147-148. The SLFH will be experienced only by trajectories intersecting or around the first model level. How is SLFH assigned to trajectories at higher altitudes?

We consider the SLHF of trajectories only if their pressure is higher than 800 hPa. This is a good assumption since CAOs, in which the SLHF takes place, are by definition convectively unstable air masses with a rapidly growing boundary layer (see also Fletcher et al. 2016).

L459: 'do not interact' Could the interaction be at upper levels by e.g. crossing above the maximum heating associated with extratropical cyclones, potentially leading to a reduction of potential vorticity?

We acknowledge that possibility and have modified the sentence the reviewer points to.

L519: The argument is plausible but given that the lower trajectories are those that exhibit PV reduction I would have thought that this is more related to the strong surface heating rather than evaporative cooling. There is no direct demonstration of evaporative cooling being responsible for negative PV in this case.

We agree with the reviewer's comment. However, we believe that surface heating and evaporative cooling both act to reduce PV.

We have addressed the subsequent technical corrections, except in instances where we've removed certain sections of the manuscript, primarily those related to GS trajectories.

Technical comments

L15: Even after reading the paper I am not quite sure what is meant by 'one-fifth of these air masses'. How are the air masses being counted or what metric is being used to make such a statement?

L48: Change 'pass through the region' to 'pass through a region'.

List of references and references in the text: Vanniére et al. (2017) and Vannière et al. (2017) are two different papers but the tilde over the first author's name can't be the symbol to distinguish them. I believe Benoît's surname is Vannière.

L107-108: 'eight neighbouring grid points' I imagine this number of grid points is resolution dependent, is it?

L110: Is it worth applying the latitude related factor to the rapidly intensifying cyclone criterion given that the cyclones in the case study do intensify at a variety of latitudes?

The correction has been implemented but not described in the previous version of the article, we included it in the revised version.

L130 and L134: As I understand it, the regime mask is based on the weather regime PV pattern. However, the mask is described in L130 but the WR PV pattern is not defined until L133-134. I'd recommend switching the sentence order so that the text becomes clearer.

L134: What is the 'cycle' referred to in this line?

L134-135: I'd suggest indicating when both NPVAs formed in the first sentence simply because otherwise the reader is left wondering why you give this information about the minor NPVA but not about the main one. Of course, this comes in the next sentence, but I think re-ordering (re-writing) these two sentences would make the text easier to read.

L163-164: I don't quite follow the additional criteria to avoid double-counting here and in section 2.2.3.

The double-counting procedure is described in detail by Madonna, et. al. 2014, but we provide an additional explanation of what it means in the revised manuscript. Overall, the goal is to avoid double counts of trajectories representing the same air mass.

L169: Are there any special considerations to make when dealing with BL trajectories in terms of the parametrisation turbulence and other boundary layer processes?

Following the suggestion of the first reviewer we removed this analysis from the manuscript.

L215: How is the start of the ascent defined? Are there any pressure fluctuations along trajectories that need to be smoothed out?

We define the beginning of ascent as the point when the pressure of the trajectory monotonically decreases, and the onset of ascent is identified at the last time step where the trajectory pressure exceeds 800 hPa. We have provided further clarification on this in the manuscript.

L245: I'm not sure I understand this sentence: "The close succession of cyclones over the North Atlantic [...] disrupted the normal progression of weather systems". I would have attributed the disruption to atmospheric blocking rather than to the succession of cyclones.

In the revised manuscript this section has been substantially modified.

L258: Should it be Fig. 1c rather than Fig. 1e?

L260: I'd recommend using the cyclone nomenclature already introduced instead of the cyclones' start dates (e.g., LE1 instead of the one from 18 February).

L262: Change 'reinforces' for 'reinforced'.

L264: By cyclone's ascending air stream, do you mean the warm conveyor belt?

The warm conveyor belt, which represents an ascent of 600 hPa within 48 hours, sets a clear benchmark for ascending airstreams within a cyclone. Many trajectories show an ascent of 500 hPa (e.g., moving from around 900 hPa to 400 hPa), but they might not meet the WCB criterium even though they are still part of the extratropical cyclone. Therefore, by saying 'ascending air stream' we use a more general criterion.

L264: Add 'the' between 'from' and 'western'.

L268: Should it be Fig. 3c rather than Fig. 1c?

L268: It might be clearer to refer to the cyclones as LE0-3.

L304 and others: In general, what is included in the article is worth noting. I'd recommend avoiding saying 'It is worth noting that…' to simply say 'Note that…' However, I recognise this is a matter of personal preference.

L301-305: Rewrite these sentences so that you first mention the separation into two subsets to then quote the percentages, noting that they are based on the strong CAO subsets.

This part of the manuscript has been modified. We hope that the revised version is more precise.

L308-309: It's not completely surprising that most trajectories are diabatically heated by 2 K or more given that by construction the trajectories are required to ascend.

While your point is valid, we still find it remarkable that a significantly higher proportion of NPVA GS trajectories undergo diabatic heating compared to NPVA nonGS trajectories. We've elaborated on this observation in the updated version of the manuscript.

L310: Delete 'the' from 'the upper-level blocks'.

L319: I understand that a reference to Madonna et al. (2014) is required here, but as it stands it's not clear why. I'd say '... of 600 hPa within 48 h (green in Fig. 4), following Madonna et al. (2004).'

L320: I think it should be 'evolution' rather than 'revolution'.

L404: Change 'ascend' for 'ascent'.

L432: Should it be Fig. 6d rather than 8d?

L435: Change 'Tab/2' for 'Tab. 2'.

L452: Change 'Fig.8, d' for 'Fig. 8d'.

L476: There is no Fig. 8f.

L486: There is only one sub-section (3.5.1) in Section 3.5. Is the header needed?

L495: Change 'level' for 'time'.

L503: Should it say 'Fig. 10a,b'?

L507: Change '...trajectories. Indicating…' for '...trajectories, indicating…'

L515: Why 36N? I would have said 40N.

L523: Change 'exceed' for 'go below'.

L539: Delete 'GS' and add 'that' between 'trajectories' and 'started'. 'GS' would be redundant if you say 'trajectories that started from the Gulf Stream'.

L590: The piece that is still missing is the decay of the block. Is it that for some reason CAOs and cyclones stop occurring and the block decays? Or is it that their WCB outflows need to comply with certain features to be able to sustain the block?

L599: Should it say '... suggested by Methven (2015)'?

Figure 1. I've always found that crosses marking parcel locations obscure quite a lot of the field underneath. Is there any other way of representing this? What about dots or perhaps you could present not so many of them?

Figure 2: The stars are very difficult to see. Perhaps this could be improved by outlining them with a black line. The caption should say what they indicate.

Figure 4: The caption should explicitly say what are the (a) and (b) panels. Is the grey background necessary? The colours for DI and CAO(>0K) are too similar to each other and therefore difficult to distinguish.

Figure 6. Caption: (left column) includes (a). It's perhaps better to say (c,e). Change (%3000 km^2) at the end of the caption to (%/3000 km^2)

Figure 7a: What are the intersections between categories, i.e. what categories are a subset of what categories and which categories should add to 100%? Add 'interval between the 90th and 10th percentiles…' The colours used for CS and rest are very similar to each other and therefore difficult to distinguish. Could you use different colours or different line thicknesses?

Figure 8: Caption: Delete 'whem'. I might have missed it but I think panel (b) was not used.

Figure 10: The order in which the variables are listed in the caption does not correspond to the panels.

---

## Author Response (AR2)

First and foremost, we want to express our sincere gratitude to the reviewer for their thorough review and the time dedicated to our article. Many of the comments provided are exceptionally valuable, and we acknowledge that various aspects of the paper required enhancement. The insightful feedback from the reviewer has significantly improved the paper. We address each of the comments below. For minor comments, unless specified otherwise, we have incorporated the reviewer's suggestions.

Major comments:

As you state in your abstract: "These findings highlight the importance of CAOs and the Gulf Stream region with their intense coupling between the ocean and atmosphere for block development and provide a mechanistic pathway linking air-sea interactions in the lower troposphere and the upper-level flow."

And in the introduction: "We investigate the potential connections between air-sea interactions over the Gulf Stream region and the formation of an upper-level ridge over western Europe using a Lagrangian perspective in a synoptic analysis."

These are the most important things shown by your work and you should not deviate from these statements (e.g., discussing dry intrusions or the PV discussion in the appendix). I have suggested several places to remove text to help keep the focus on your aims, which will also make the paper much more concise.

We thank the reviewer for their advice on how to better keep the main focus on the dynamic link between air-sea interaction and blocking over Europe. We have followed most of the suggestions and hope to even better streamline the storyline, as detailed in the answers to the specific comments below. However, we have not fully removed some of the side tracks. This is because these are aspects which we received a lot of interest from the community when presenting the work on workshops and conferences. We still think with the corrections made these statements are not distracting from the main focus.

My largest major concern, however, is around Figs 9 and 10. The figures are good, but they are not focussed on the main synoptic systems you describe in the text up to that point (i.e., L1 and L2). If they were re-produced for those cyclones, then I think the paper would read very well.

Thank you for the hint. Initially, we selected the time steps and cyclones shown, which best illustrate the general characteristics of the "handover mechanism". However, we agree that it will be easier if cyclones L1 and L2 are shown and changed the figures accordingly.

Again, I thoroughly commend the authors for their revisions so far and I am very supportive of this work being published once the major concerns have been addressed.

Major comments:

Lines 172-175, Figure 2: Your segregation into "inflow", "ascent" and "outflow" is confusing here, and is not truly reflective of what is going on. Parcels below 800 hPa (i.e., nearer to the surface) are not just "inflow" parcels. They are either undergoing modification or forming the inflow. Given that the inflow is only happening at the domain boundary (i.e., across the 800 hPa surface), most of the parcels will be undergoing "modification" rather than forming the inflow. I suggest you change the naming to "modification" layer as this is more reflective of what is going on (and what you are interested in). You can state that these air parcels would be in the boundary layer, over the Gulf Stream, and therefore have the potential to be modified by the high SSTs in that region. You can then refer to the 800 – 400 hPa layer as the "inflow and ascent" layer. Using the Binder et al. (2020) paper as justification for your level choices is fine, but I think using "modification" for the lowest layer is much better language and fits in better with all your arguments. You do not have to use the same language Binder et al. (2020) as your study is focussing on something else (you can just state, "called the inflow layer in Binder et al. (2020)"). Binder et al. (2020) were looking at warm conveyor belts, whereas the modification of air parcels before entering the warm conveyor belt is important here so using different wording is fully justified.

In summary: I would change 172 – 175 and state:
"An advantage of using the trajectory analysis is that we can identify where the NPVA trajectories are at different times throughout the February 2019 blocking event. The position of the air parcels can be grouped into different layers in a similar manner to that employed by Binder et al. (2020). The layers are defined here as:
1. The modification layer (p ≥ 800 hPa, called the "inflow" layer in Binder et al., 2020) where parcels are being modified by the underlying ocean.
2. The inflow and ascent layer (800 ≤ p ≤ 400 hPa) where parcels are flowing in to the WCB and then undergoing ascent.
3. The outflow layer (p ≤ 400 hPa) where parcels stop rising and diverge near to the tropopause.
The layers will help to elucidate the location and transfer of air parcels that are important to the February block development and maintenance. "
At the moment lines 172 – 175 appear to just be "tagged onto the end of a section" whereas stating how/what/why you are doing it clearly (in a similar way to above) means the description of Fig. 2 in section 3 makes a lot more sense. I really liked Fig. 2, once I'd worked out what was going on.

We thank the reviewer for their insightful comment and understand the perspective from which it was offered. However, after careful consideration, we have decided to continue using the established terms 'inflow', 'ascent', and 'outflow'. Our decision to adhere to these well-established terms is informed by their widespread acceptance and use in studies on ascending airstreams, e.g. Binder et al. 2020, Schäfler et al. 2014, Pickl et al. 2023, and Quinting et al. 2022 and others. We consider air streams that ascend from the lower to the upper troposphere with a pressure difference of at least 500 hPa. A substantial fraction of these air streams

fulfilled the criteria of warm conveyor belts which were the focus of earlier work on midlatitude ascending air streams. As the mechanisms facilitating ascent, latent heat release, and diabatic outflow are essential, we think it is justified to adopt previous terminology for our analysis of airstreams interacting with the Gulf Stream and ascending to the upper level NPVA. We believe that introducing new terms in place of these familiar ones could lead to confusion within the scientific community rather than aiding clarity in our paper. Regarding the $800 \leq p \leq 400$ hPa layer, we respectfully disagree with the suggestion to label it as an 'inflow and ascent' layer. Our results, as clearly demonstrated in Figure 4, indicate that the pressure drop below 800 hPa is associated with a rapid moisture loss, indicative of ascent rather than inflow. Recognizing the importance of clarity, we have included additional explanations in our manuscript to elucidate this distinction more effectively and to better explain what we mean with the inflow layer. Also we now explicitly state that air mass modification occurs in the inflow layer and later the air converges into the ascent. These textual changes have been made in lines 170-180 as suggested.

L222 – 224: Following on from above, you state:
"On 18 February, the NPVA GS trajectories were found to be in their ascent phase, distinctly spread over the western North Atlantic, as shown by the red crosses in Fig.2a. Yet, the bulk of air parcels that later ascend into the blocking region remained in the lower troposphere. These air parcels were predominantly observed in regions influenced by CAOs in the western and central parts of the North Atlantic…"

The wording is confusing here because it draws you towards thinking that all the air parcels are associated with the CAO, which they're not (and you don't say they are). This is where using the word "modification" could really help. How about changing the wording to:
"On 18 February, the NPVA GS trajectories were found to be in their ascent phase, distinctly spread over the western North Atlantic, as shown by the red crosses in Fig.2a. Yet, the bulk of air parcels that later ascend into the blocking region remained in the modification layer in the lower troposphere ($p \geq 800$ hPa). These air parcels were predominantly observed in regions influenced by CAOs in the western and central parts of the North Atlantic (green crosses Fig. 2b)…"

In line with our previous response, we have chosen to retain the current naming convention for the layers. However, in response to the reviewer's comment, we have clarified in our manuscript that these air masses are distinctly influenced by the processes at the ocean-atmosphere interface (lines 235-240).

Section 3.2: This section is really interesting, and you have made a fantastic effort to revise it. I'm also happy with your arguments overall. What would make this section much better would be to write the text in the order that the figures are written. You currently describe Fig 4a, then 4e, then 4f, then 4c and 4d, which leads the text to read in a disordered way. Either adjust your figure order or re-order the text to follow the figures in sequence. This will make it much easier to

read. Your arguments are good here, but the non-sequential ordering of the text and figures hides it somewhat.

We are grateful to the reviewer for their insightful feedback. In response, we have corrected both the figure (Fig. 4) and the corresponding text to better align the text and figure.

L314-325: The DI paragraph is unnecessary. You do not state this as something you are going to look at in either the abstract or introduction. The paragraph does not fit in with the focus of the paper and unnecessarily lengthens it. While this could be an interesting line of research to follow (as you say), it adds nothing to this paper and only detracts from the already good hypotheses and evidence presented.

We fully appreciate the reviewer's concerns regarding this paragraph. However, after careful consideration, we have decided to retain the short paragraph in question. We believe this section offers valuable information to the scientific community, a perspective supported by discussions at various scientific meetings and recent publications e.g. Demirdjian et. al.2023. The link between DI, CAO, and DH along the trajectories has captured the interest of researchers during our presentations of the case study results at conferences. Therefore, we feel that its inclusion adds value to our study, and at the same time doesn't create much confusion. We now clearly state that this paragraph is a sidetrack (lines 330-340).

Figs 9 and 10 and the end of Section 3.3 (L416 to L459): After all the great description of the processes happening around L1 and L2 in the preceding text, why have you decided to plot a new set of lows (l0, l1, l2) with only a passing reference to L1 in Figure 10? Furthermore, you state in the introduction (L85-86), "This event brought record-breaking winter "heat" to Western Europe and was accompanied by a series of upstream, rapidly intensifying cyclones." Moreover, you also state at L370 that "…NPVA GS air parcels coincide with the period when cyclones L1 and L2 are present in the North Atlantic…". You have shown that cyclones L1 and L2 are very important so the focus should be (and stay) on the rapidly intensifying L1 and L2 cyclones. If this analysis had been performed on the L1 – L2 sequence, then it would have supported the rest of the arguments in the paper. I find the choice here very confusing. If the focus of your paper had been the sequence of lows plotted in Figs. 9 and 10 then it would make sense here, but that is not the case. You need to either re-write everything before Section 3.3 (and the introduction) and focus on the events you plot in Figures 9 and 10 (I don't recommend this, but it is an option) or change Figures 9 and 10 to look at what happens around L1 and L2 (recommended). I can imagine the L1 – L2 (and beyond) sequence of events would also show the same processes you plot in Figures 9 and 10.

We concur with the reviewer's suggestion regarding this matter. We had chosen the initial time step to more effectively illustrate the mechanism of moisture uptake in 'recirculating trajectories'. However, upon reflection and considering the reviewer's input, we recognize the importance of focusing on the L1 and L2

cyclones. This approach aligns more closely with the trajectory analysis we have presented throughout the article. Therefore, we have updated the paragraph in Section 3.3 and moved the previously used figure to the Appendix. We chose to keep this figure in the Appendix because it clearly shows the recirculation in some of the case study trajectories. We think that these changes will improve the clarity and usefulness of the section, and we look forward to the reviewer's feedback on these modifications.

L471-490 and Fig 11: I like this schematic and description, but it needs to reflect the changes I've suggested for Figs 9 and 10 (which is why I have put it in the major revisions section). If Fig 11 is re-made to focus on L1 and L2 and lines 471-490 re-worded to reflect the changes to Fig 11 then I think this will work very well.

Following the same rationale as in our response to the previous comment, we have decided to incorporate the reviewer's suggestion, as we agree that it integrates well with the analysis presented in our paper.

Minor points:

Line 6: Change "responsible for" to "contributing to" as you show their contribution to the block, not their responsibility for it.
Line 20: Change "for blocking" to "contributing to blocking" – again, you don't show that the coupling causes blocking rather that it contributes to the blocking event.
L24: Change to "… associated surface high-pressure system…"
L47-49: Remove the sentence beginning "The air masses that undergo…" as this implies that all air masses must go through this process, which is not necessarily true. It is just one of the processes that air masses can undergo to gain moisture before ascending. This is a case study, and not "all events" are considered, therefore what might be true in this case might not be true always.
L63: Change "can regulate cyclone formation and strength" to "are important for cyclone development" – maybe they can regulate formation and strength, but I think you're more concerned with their importance, which is why I've suggested the word change.
L77: Change "ascent" to "ascend".
L107: remove "the" before "sea level pressure".
L133-136: Move to L215 – see comment for L215-218 below.
Comment addressed below.
L159: Better description now, but this is still ambiguous – "equidistant grid points of size 100x100 km." Do you mean "the domain of the grid points is 100 x 100 km" or "each grid point is 100 km away from each of its neighbours"? I'm guessing the second option; in which case it is probably better to say "equidistant grid points with a separation of 100 km between each point".
L171: Remove the words "and their subsets (NPVA GS and NPVA nonGS)" as you haven't defined these yet (I have suggested another edit below to account for this change).
L178: Change, "…over the Gulf Stream in the lower troposphere. We define the…"" to "…over the Gulf Stream in the lower troposphere (NPVA GS, see Table

1). To identify the NPVA GS trajectories, we…".

L184-187: Remove "In the following, we refer to those trajectories as 'NPVA GS trajectories (Tab 1)" and move the next sentence up to join the paragraph that finishes on L184. Start L187 by saying, "The NPVA GS and NPVA nonGS trajectories are split into their inflow, ascent and outflow stages (as described in Section 2.2.1). For each trajectory within the NPVA GS…".

L215-218: Remove these lines up to "…upper-troposphere NPVA" and replace it with the words between L133 – 136, which provide much better introductory sentences.

We acknowledge and understand the rationale behind the reviewer's suggestion concerning the paragraph's structure. However, after thoughtful consideration, we have chosen to keep its original format. Our concern is that the integration of text from lines L133-136 into the existing paragraph, as suggested, might lead to confusion. The paragraph beginning at line 215 is centered on the NPVA over the North Atlantic, and introducing information about the NPVA's lifecycle at this point could disrupt the focus. Therefore, we believe that maintaining the current structure of the paragraph ensures a clearer and more effective communication of the intended information.

L222: Remove "distinctly".
L226: Change "very intensive" to "very intense".
L225-226: "Those CAOs occurred behind the very intense cyclone L0 (Fig 3a)…" – really? The cyclone near Canada in the Labrador Sea seems to be a much more likely candidate for causing this CAO. Even the analysis in Fig 3a seems to place L0 too far to the east to cause the CAO. Please check this.
We appreciate the reviewer's feedback and have made the necessary edits to the text as suggested, mentioning the low-pressure system over Labrador Sea.

L233: Change from l1.1 and l1.2 to L1.1 and L1.2 (i.e. upper case). The ".1" and ".2" are sufficient for showing these are supplementary lows. Using the lower case "l" makes it look like 11.1 and 11.2.
L233: "The transit of this cyclone…" which cyclone? You mention L1, L1.1 and L1.2 so you need to be clear which one you mean (I assume it is L1).

L239-240: Remove the sentence starting, "The green crosses in 2f…".
We value the reviewer's suggestion; however, we do not understand the rationale behind the idea of removing those sentences. We believe they provide the necessary context, making it easier for readers to follow the text and what is depicted in the figure.

L241-242: Remove the sentence starting, "This structure not only…"
Similarly to our response to the previous comment, we are uncertain about the reasoning for the suggested removal of these sentences. After careful consideration, we have chosen to retain them in the text.
L242-243: You state that "… the L2 cyclone propagated into the region of high surface fluxes…" but you do not show it. As the cyclone propagates, it would displace the coldest air around the cyclone on its northern flank and draw up the

heavily modified former CAO (now warm sector) air over the region. The way you have worded it, it reads like you're saying the cyclone would propagate over the CAO, whereas the region would be subject to deformation around the cyclone. More likely is that the cyclone propagates on the boundary between the residual cold air from the CAO and the warmer, heavily modified air to the south. If you keep this statement then you need to show it happening, or I suggest removing this sentence.

We appreciate the reviewer's suggestion and understand the reasoning behind it. However, after careful consideration of the reviewer's comments, we opted to revise the sentence rather than remove it. This decision is informed by further analysis, enhanced by the reviewer's feedback, which reveals that the moisture sources for trajectories ascending with cyclone L2 predominantly originate from regions impacted by those CAOs. Additionally, this characteristic behavior of the cyclone aligns with the findings presented in the research works by Papritz et al. 2021 and Dacre et al. 2019, which are frequently referenced in our study. The sentence now reads: *It is important to highlight the fact as cyclone L2 propagated, it traveled into the region where the air in the lower troposphere has been heavily modified due to the surface fluxes that occurred in the wake of cyclone L1.*

L247: Remove "predominantly".
L249-251: Remove sentences starting, "Unlike cyclones L1 and L2… and black crosses in Fig 2g." There is a lot of conjecture here and it detracts from the results you are describing.
L254-258: Remove these lines as you don't look at these cyclones in any detail (apart from maybe Figs 9 and 10, but I don't think you should use these cyclones to produce Figs 9 and 10 – see major points).
Addressing two comments above: We appreciate the reviewer's perspective on this matter. However, we respectfully maintain a different view. As cyclones L3 and L4 are prominently featured in our figures, we believe it is essential to mention and provide explanations for them in the text to ensure clarity and completeness of our study.

L285: I think it should say, "within a 48-hour interval".
L291: Start a new paragraph here.
L306: change "an interplay between the block and preceding cyclones" to "the air transport through the preceding cyclones has influenced the block." There is no "interplay" as the cyclones affect the block but the block does not impact the cyclones as they have already gone by that point.
L313: Remove reference to Appendix A (due to removal of Appendix A).
L314-325: Just re-iterating to remove these lines as they do not contribute to this work nor support its conclusions.
This comment has been addressed in the response to Reviewer's major comments.
L338: No need for brackets around the Sodemann et al. reference.
L341-345: Suggest re-wording to the following so it focuses on the contents of the table:
"First, we will focus on the timing and spatial distribution of moisture uptakes of the

trajectories. NPVA GS trajectories, on average, accumulate moisture around 3.5 days before their ascent (Tab.3) with the most significant portion (60%) acquired in the first five days. In comparison, NPVA nonGS trajectories start collecting moisture about 3.8 days prior to ascent (Tab.3), with 48% of uptakes taking place within the first five days."

L355-356: Remove "the majority of the moisture originates from regions relatively close to the block (Fig. 3e). In particular" so that the sentence reads (and gets directly to the important point):

"Our moisture source identification methodology indicates that 80-90% of the uptakes…"

L365: change "when the cold" to "when cold".

L370: Remove "and L2" from the sentence. The reason for this is that L2 forms when the SLHF is very high i.e., it is high because of the CAO following L1 and L2 has nothing to do with that CAO at the time you're looking at.

L380: Start a new paragraph after "NPVA.", which should then continue as the same paragraph through L383.

L394-395: Change "Cyclone L2 traverses and strengthens within the region marked by a strong CAO left in the wake of L1 (Fig.2b and d) ). This moisture-rich air potentially gets" to "Cyclone L2, traverses to the north of this heavily modified, moisture-rich air that then gets". The way it is currently worded, it reads like the cyclone propagates into the CAO, which is very unlikely given the CAO will have lost a lot of its signature. More likely is the increase in baroclinicity between the heavily modified air and the residual CAO acts as the boundary along which the cyclone propagates. My point here also applies to the point I made about L242-243 i.e., why not just show L2 propagating through the region around ~60W and we can see where the residual CAO, modified CAO and low centre are in relation to each other?

As suggested by the reviewer, we have updated the text accordingly. Moreover, as shown in Fig. 3, cyclone L2 moves into the region around ~60 W. This is further illustrated in Figs. 9 and 10, where it's evident that the moisture uptakes for trajectories ascending with this cyclone occur within the region previously affected by CAO, induced by cyclone L1. We acknowledge that this aspect may not have been clearly articulated in the original text, and have therefore made necessary modifications (lines 410 - 415). These revisions are intended to present the information more clearly, and we trust that they now effectively address the reviewer's concerns in a more comprehensive manner.

L402: "exemplary trajectory discussed before" – which example is this? Why not just say "NPVA GS trajectories"? I even think you could remove this sentence entirely without any impact on the paper.

We appreciate the reviewer's insightful comment and recognize that certain elements of our initial presentation might not have been sufficiently clear. To address this, we wish to emphasize that our analysis encompasses all time steps of the case study, available in the Supplementary Material. In the main text, we illustrate this analysis through a single example from a specific time step. This example is chosen to clearly show how the trajectories' ascent and moisture

uptake are affected by cyclones L1 and L2. We have made revisions to the text to ensure this information is conveyed more clearly and effectively.

L407-415: These can be removed along with Figure 8 as they don't add anything to the paper. You have plenty of other good descriptions of the processes and this figure and explanation are not needed. It will make the paper more concise.

We respectfully differ in opinion from the reviewer regarding Fig. 8. We believe that it offers support for the prevalence of the hand over mechanism in NPVA GS trajectories. Additionally, we are concerned that omitting these statistics might prompt questions from readers about the distribution of time differences between the ascent and uptake phases. We value the reviewer's perspective and have carefully considered this aspect in formulating our response.

L464: "disproportional role" – I don't think you have shown this. You show evidence of their involvement, not the proportion of the blocking circulation that they are contributing to. You would likely need to do some sort of PV surgery method to show the roles of the different air parcels from GS and nonGS sources. I do not think such PV surgery is necessary in your paper, however as you show GS trajectories clearly do make it into the block all you need to say is "we show evidence that they have a role in maintaining or enhancing…".
We edit the text according to reviewer's suggestions.
L417: Remove "often".
L471: change "… one cyclone ascends…" to "one cyclone can ascend…" – this is a case study and so you don't show this always happening.
L475: Change "The most robust…" to "The largest…"
L504-505: "Therefore, for the air masses that ascend into the block during the extratropical cyclones of February 2019 in the North Atlantic, these distant moisture sources seem less influential" – even though these air masses contribute most to the NPVA region? You are overstating what you show here. The paragraph reads well up to this point and is consistent with your main results. You should remove the sentence above.
L520-523: Remove the sentence starting "Consequently, we hypothesize…" – you don't discuss the termination stage and so this statement leaves me thinking "how do you know this?". You either need to show it (not recommended) or remove it. The paper does not need this statement about the decay stage as it is not the focus.

We value the reviewer's insights and have chosen to act on their recommendation by omitting the specified sentence.

L527: Change "significance of CAOs" to "significance of the modification of CAOs" – it is the modified CAO air that contributes to the block. It is no longer "cold air" nor an "outbreak" by the time it starts to ascend. The air must be modified to do this, which you paper clearly shows. Just stick with describing what you have shown.
End of the conclusions: Your future work sounds very interesting!

Appendix A: Please remove this. It doesn't add to your analysis and makes the paper longer. You have a lot of excellent analysis that clearly show all the relevant processes. This section does not fit in with what you've written and makes the paper longer than it needs to be.

We recognize the reviewer's concerns about including this Appendix and understand their viewpoint regarding its potential lack of necessity. While we agree that its content may not be critical for the main body of the article, we have chosen to keep it in the Appendix. This decision is based on its relevance to ongoing research, of which we are aware, being conducted by other groups. Additionally, the topic has sparked numerous discussions during the preparation of our article, leading us to anticipate that these results are expected to be published. Placing this analysis in the Appendix seems the most suitable approach, as it allows us to share this important information without disrupting the main text's structure and focus.